# TreeVI: Reparameterizable Tree-structured Variational Inference for Instance-level Correlation Capturing

Junxi Xiao[1]    Qinliang Su[1,2*]

[1]School of Computer Science and Engineering, Sun Yat-sen University, Guangzhou, China
[2]Guangdong Key Laboratory of Big Data Analysis and Processing, Guangzhou, China
xiaojx7@mail2.sysu.edu.cn suqliang@mail.sysu.edu.cn

## Abstract

Mean-field variational inference (VI) is computationally scalable, but its highly-demanding independence requirement hinders it from being applied to wider scenarios. Although many VI methods that take correlation into account have been proposed, these methods generally are not scalable enough to capture the correlation among data instances, which often arises in applications involving graphs or explicit constraints among instances. In this paper, we developed the Tree-structured Variational Inference (TreeVI)[2], which uses a tree to capture the correlation among latent variables in the posterior. We show that samples from the tree-structured posterior can be reparameterized efficiently and parallelly, making its training cost just 2 or 3 times that of VI under the mean-field assumption. To capture correlation with more complicated structure, the TreeVI is further extended to the multiple-tree case. Furthermore, we show that the underlying tree structure can be automatically learned from training data. With experiments on synthetic datasets, constrained clustering, user matching and link prediction, we demonstrate the efficacy of TreeVI in capturing instance-level correlation in posteriors and enhancing the performance of downstream applications.

## 1 Introduction

Variational inference is a probabilistic method that is widely used for approximating the exact posterior in latent-variable models $p(\mathbf{X}, \mathbf{Z}) = p(\mathbf{X}|\mathbf{Z})p(\mathbf{Z})$, with $\mathbf{X}$ and $\mathbf{Z}$ being the observed data and latent variable, respectively. When not considering the existence of any relation among instances, we can write the model into a factorized form as $p(\mathbf{X}, \mathbf{Z}) = \prod_i p(\mathbf{x}_i|\mathbf{z}_i)p(\mathbf{z}_i)$, where $\mathbf{x}_i$ and $\mathbf{z}_i$ denote the $i$-th data instance and latent variable. Thanks to the factorized form, we can reasonably use the mean-field posterior $q(\mathbf{Z}|\mathbf{X}) = \prod_i q(\mathbf{z}_i|\mathbf{x}_i)$ for model inference and training. However, there are also many circumstances, under which complex relations among instances exist. For instances, in graph-structured data, VGAE [19] uses two instances' latent representations to model the probability of existing an edge $e_{ij}$ between them via $p(e_{ij}|\mathbf{z}_i, \mathbf{z}_j)$, which, obviously, makes all instances coupled together. Also, in the constrained clustering, DC-GMM [24] designs a prior $p(\mathbf{c}; \mathbf{W})$ to specify the assignment constraints and then uses it to construct the prior distribution $p(\mathbf{Z}, \mathbf{c}) = p(\mathbf{c}; \mathbf{W}) \prod_i p(\mathbf{z}_i|c_i)$, which will also cause all $\mathbf{z}_i$ to be correlated with each other. Obviously, under these circumstances, due to the existence of correlation among instances, it is unreasonable to assume a factorized form for the posterior anymore.

---

[*]Corresponding author.
[2]Code is available at: https://github.com/Mephestopheles/TreeVI.

38th Conference on Neural Information Processing Systems (NeurIPS 2024).

There has been many work on incorporating correlation structures into posterior approximation in variational inference recently [23, 31, 43], and they are mainly focused on modeling correlation structure among different dimensions within each latent variable. However, the amount of latent dimensions is usually no more than a few hundred, at a completely different level from the size of dataset which might be as large as millions. Therefore, these variational inference methods modeling correlations among dimensions cannot be leveraged to capture instance-level correlation especially for large datasets. To investigate correlations among different instances, Tang et al. [36] recently used a tree structure over latent variables to take instance-level correlations into consideration, but only capable of capturing correlations between immediately connected nodes.

In this work we propose a novel posterior approximation for variational inference, the Tree-structured Variational Inference (TreeVI), that models the latent correlation structure with a tree-structure-induced distribution and enables modeling high-order correlations between non-adjacent latent variables. Our method provides a matrix-form reparametreization for tree-structured correlated latents based on ancestral sampling, that is scalable to large datasets with computational complexity comparable to mean-field variational inference methods. To better capture structural correlation, we also generalize TreeVI to multiple trees, and propose a continuous optimization algorithm to stochastically learn a theoretically correlation-rich tree or mixture-of-trees structure.

## 2 Tree-structured Variational Inference

Consider a latent-variable model $p_{\boldsymbol{\theta}}(\mathbf{X}, \mathbf{Z}) = p_{\boldsymbol{\theta}}(\mathbf{X}|\mathbf{Z})p(\mathbf{Z})$, where $\mathbf{X} = [\mathbf{x}_1^\top, \mathbf{x}_2^\top, \cdots, \mathbf{x}_N^\top]$ and $\mathbf{Z} = [\mathbf{z}_1^\top, \mathbf{z}_2^\top, \cdots, \mathbf{z}_N^\top]$; and $\mathbf{x}_i$ and $\mathbf{z}_i \in \mathbb{R}^D$ denote the $i$-th data instance and its corresponding latent variable, respectively. Given the latent-variable model $p_{\boldsymbol{\theta}}(\mathbf{X}, \mathbf{Z})$, variational inference aims to find a distribution that is closet to the true posterior $p_{\boldsymbol{\theta}}(\mathbf{Z}|\mathbf{X})$ from a distribution family $\mathcal{Q}$, which is achieved by maximizing the evidence lower bound (ELBO)

$$\mathcal{L}(\boldsymbol{\theta}, \boldsymbol{\phi}) = \mathbb{E}_{\mathbf{Z} \sim q_{\boldsymbol{\phi}}(\mathbf{Z}|\mathbf{X})} \left[ \log p_{\boldsymbol{\theta}}(\mathbf{X}, \mathbf{Z}) - \log q_{\boldsymbol{\phi}}(\mathbf{Z}|\mathbf{X}) \right], \tag{1}$$

where $q_{\boldsymbol{\phi}}(\mathbf{Z}|\mathbf{X}) \in \mathcal{Q}$ denotes the approximate posterior parameterized by $\boldsymbol{\phi}$. In most of current works, the latent-variable model is assumed to take a factorized form as $p_{\boldsymbol{\theta}}(\mathbf{X}, \mathbf{Z}) = \prod_{i=1}^N p_{\boldsymbol{\theta}}(\mathbf{x}_i|\mathbf{z}_i)p(\mathbf{z}_i)$ due to the observed independence among data instances. With the factorized form for the model, it can be seen that the true posterior $p_{\boldsymbol{\theta}}(\mathbf{Z}|\mathbf{X})$ also takes a factorized form, hence it is reasonable to choose a fully factorized form $q_{\boldsymbol{\phi}}(\mathbf{Z}|\mathbf{X}) = \prod_{i=1}^N q_{\boldsymbol{\phi}}(\mathbf{z}_i|\mathbf{x}_i)$ for the approximate posterior. However, for applications like constrained clustering with generative model and generative modeling on graph data, we often need to impose some constraints among the latent variables $\{\mathbf{z}_i\}_{i=1}^N$ by using a correlated prior distribution $p(\mathbf{Z})$ with $p(\mathbf{Z}) \neq \prod_{i=1}^N p(\mathbf{z}_i)$, or use several latent variables $\mathbf{z}_j$ to be responsible for generating a data instance $\mathbf{x}_i$ (*e.g.*, $\mathbf{x}_i$ representing an edge), which will result in $p_{\boldsymbol{\theta}}(\mathbf{X}|\mathbf{Z}) \neq \prod_{i=1}^N p_{\boldsymbol{\theta}}(\mathbf{x}_i|\mathbf{z}_i)$. Obviously, under these scenarios, latent variables $\{\mathbf{z}_i\}_{i=1}^N$ from the true posterior $p_{\boldsymbol{\theta}}(\mathbf{Z}|\mathbf{X})$ are not independent anymore. Thus, if a factorized posterior $q_{\boldsymbol{\phi}}(\mathbf{Z}|\mathbf{X}) = \prod_{i=1}^N q_{\boldsymbol{\phi}}(\mathbf{z}_i|\mathbf{x}_i)$ is still employed to approximate the true posterior, the model will lose the ability to capture the correlations among data instances. To alleviate the deviation caused by mean-field approximation, we propose to approximate the correlation structure with a tree or a mixture-of-tree structure as shown in Fig. 1, respectively.

### 2.1 TreeVI: Variational Inference under a Single Tree

To capture the correlations among latent variables $\{\mathbf{z}_i\}_{i=1}^N$ in the posterior distribution, a multi-variate normal distribution $q(\mathbf{Z}|\mathbf{X}) = \mathcal{N}(\mathbf{Z}; \boldsymbol{\mu}_\mathbf{z}, \boldsymbol{\Sigma}_\mathbf{z})$ can be used, where $\boldsymbol{\mu}_\mathbf{z} = [\boldsymbol{\mu}_1^\top, \cdots, \boldsymbol{\mu}_N^\top]^\top \in \mathbb{R}^{ND}$ is the mean and $\boldsymbol{\Sigma}_\mathbf{z} = \mathrm{diag}(\boldsymbol{\sigma}_\mathbf{z})\mathbf{R}\,\mathrm{diag}(\boldsymbol{\sigma}_\mathbf{z})$ is the $ND \times ND$ covariance matrix; $\boldsymbol{\sigma}_\mathbf{z} = [\boldsymbol{\sigma}_1^\top, \cdots, \boldsymbol{\sigma}_N^\top]^\top \in \mathbb{R}^{ND}$ is the standard deviation; and $\mathbf{R}$ means the correlation matrix, with its $(i, j)$-th block taking the form

$$[\mathbf{R}]_{ij} = \mathrm{diag}(\boldsymbol{\gamma}_{ij}); \tag{2}$$

$\boldsymbol{\gamma}_{ij} \in (-1, 1)^D$ for $i \neq j$ denotes the correlation strength between latent variable $\mathbf{z}_i$ and $\mathbf{z}_j$ and $\boldsymbol{\gamma}_{ii} \triangleq \mathbf{1}_D$ for $i \in \mathcal{V} \triangleq \{1, 2, \cdots, N\}$. Here, the diagonal structure assumed for the $(i, j)$-th block $[\mathbf{R}]_{ij}$ implies that we are only interested in capturing the correlation among different variables, without considering the correlation of different dimensions within one variable. Actually, due to the

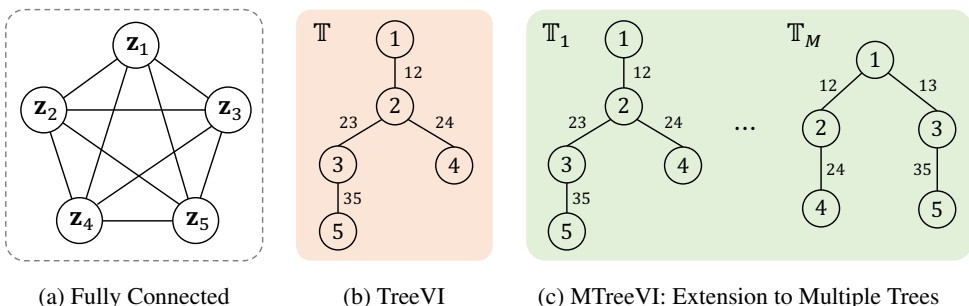

|   |   |   |
|---|---|---|
| (a) Fully Connected | (b) TreeVI | (c) MTreeVI: Extension to Multiple Trees |

Figure 1: Comparison between the fully-connected correlation structure and approximation by single and multiple tree-structured variational inference

relatively low dimensions of a latent variable $\mathbf{z}_i$, the correlation within a variable can be captured by combining with existing VI methods, but we here omit the modeling of correlation among dimensions for simplicity. In the following context, the diagonal operators are omitted for conciseness.

A crucial part of the variational inference is to determine appropriate values for the correlation parameters $\mathbf{\Gamma}^0 = \{\boldsymbol{\gamma}_{ij} : i \neq j \in \mathcal{V}\}$. For a general multivariate normal distribution, all correlation parameters in $\mathbf{\Gamma}^0$ are learnable, which admits a highly expressive correlation structure, but also resulting in high computational complexity. The widely-used mean-field approximation reduces the cost by assuming an independence structure within any pair of latent variables, *i.e.*, $\boldsymbol{\gamma}_{ij} = \mathbf{0}_D$ for any $i \neq j$, completely ignoring the correlation among latent variables. In order to achieve a balance between expressiveness and complexity, we propose to approximate the fully-connected correlation structure with a tree structure, introducing our TreeVI, namely tree-structured variational inference.

The idea of tree-structured variational inference is to impose a tree correlation structure $\mathbb{T} = (\mathcal{V}, \mathcal{E})$ among latent variables, as illustrated in the quin-variate example in Fig. 1b. In our TreeVI, given a tree structure $\mathbb{T} = (\mathcal{V}, \mathcal{E})$, we only specify the correlation parameters $\boldsymbol{\gamma}_{ij}$ for latent variables $(\mathbf{z}_i, \mathbf{z}_j)$ adjacent on the tree, and for any non-adjacent latent variables $(\mathbf{z}_i, \mathbf{z}_j)$, the correlation between then are directly computed as the multiplication of correlation parameters along $\mathbb{P}_{i \to j}$, that is

$$\tilde{\boldsymbol{\gamma}}_{ij} = \prod_{(s,t) \in \mathbb{P}_{i \to j}} \boldsymbol{\gamma}_{st}, \quad (i,j) \notin \mathcal{E}, \tag{3}$$

where $\mathbb{P}_{i \to j}$ denotes the path connecting the latent variable $\mathbf{z}_i$ to $\mathbf{z}_j$ on the tree $\mathbb{T}$, and the vector multiplication is assigned as Hadamard product. Therefore, only the correlation strengths w.r.t. the edges of the tree $\mathbb{T}$ are learnable, which can be denoted as $\mathbf{\Gamma}^{\mathbb{T}} = \{\boldsymbol{\gamma}_{ij} : (i,j) \in \mathcal{E}\} \subset \mathbf{\Gamma}^0$. The parameterization leads to a correlation matrix $[\mathbf{R}^{(\mathbb{T})}]_{ij} = \tilde{\boldsymbol{\gamma}}_{ij}$ that is totally determined by parameters in $\mathbf{\Gamma}^{\mathbb{T}}$, with $\tilde{\boldsymbol{\gamma}}_{ii} = \mathbf{1}_D$ for $i \in \mathcal{V}$. For example, for the quin-variate tree structure in Fig. 1b, the correlation parameter between $\mathbf{z}_1$ and $\mathbf{z}_5$ is fixed as $\tilde{\boldsymbol{\gamma}}_{15} = \boldsymbol{\gamma}_{12} \odot \boldsymbol{\gamma}_{23} \odot \boldsymbol{\gamma}_{35}$.

With the tree-structured correlation defined above, it can be easily shown that the distribution $q_{\boldsymbol{\phi}}^{\mathbb{T}}(\mathbf{Z}|\mathbf{X})$ forms a Markov random field defined by the tree $\mathbb{T}$ [3], and its joint probability can be expressed as

$$q_{\boldsymbol{\phi}}^{\mathbb{T}}(\mathbf{Z}|\mathbf{X}) = \prod_{i \in \mathcal{V}} q_{\boldsymbol{\phi}}(\mathbf{z}_i|\mathbf{x}_i) \prod_{(i,j) \in \mathcal{E}} \frac{q_{\boldsymbol{\phi}}(\mathbf{z}_i, \mathbf{z}_j|\mathbf{x}_i, \mathbf{x}_j)}{q_{\boldsymbol{\phi}}(\mathbf{z}_i|\mathbf{x}_i) q_{\boldsymbol{\phi}}(\mathbf{z}_j|\mathbf{x}_j)}, \tag{4}$$

where the marginal distribution $q_{\boldsymbol{\phi}}(\mathbf{z}_i|\mathbf{x}_i) = \mathcal{N}(\mathbf{z}_i; \boldsymbol{\mu}_i, \mathrm{diag}(\boldsymbol{\sigma}_i^2))$, and pairwise marginal distribution for adjacent variables $(i,j) \in \mathcal{E}$ is $q(\mathbf{z}_i, \mathbf{z}_j|\mathbf{x}_i, \mathbf{x}_j) = \mathcal{N}(\mathbf{z}_i, \mathbf{z}_j; \boldsymbol{\mu}_{ij}, \mathbf{\Sigma}_{ij})$; and the mean $\boldsymbol{\mu}_{ij} = [\boldsymbol{\mu}_i^\top, \boldsymbol{\mu}_j^\top]^\top$ and covariance matrix

$$\mathbf{\Sigma}_{ij} = \begin{bmatrix} \boldsymbol{\sigma}_i \odot \boldsymbol{\sigma}_i & \boldsymbol{\gamma}_{ij} \odot \boldsymbol{\sigma}_i \odot \boldsymbol{\sigma}_j \\ \boldsymbol{\gamma}_{ij} \odot \boldsymbol{\sigma}_i \odot \boldsymbol{\sigma}_j & \boldsymbol{\sigma}_j \odot \boldsymbol{\sigma}_j \end{bmatrix}; \tag{5}$$

$\boldsymbol{\gamma}_{ij}$ is the correlation parameter between two adjacent variables. Furthermore, we can also see that the distribution $q_{\boldsymbol{\phi}}^{\mathbb{T}}(\mathbf{Z}|\mathbf{X})$ can be represented by an acyclic Bayesian network of the form

$$q_{\boldsymbol{\phi}}^{\mathbb{T}}(\mathbf{Z}|\mathbf{X}) = q_{\boldsymbol{\phi}}(\mathbf{z}_1) \prod_{(i,j) \in \mathcal{E}} q_{\boldsymbol{\phi}}(\mathbf{z}_j|\mathbf{z}_i), \tag{6}$$

where the conditioning on data $\mathbf{x}_i$ is omitted for conciseness; $\mathbf{z}_1$ is assumed to be the root node; the conditional distribution of latent variables with respect to an edge $(i, j) \in \mathcal{E}$ is

$$q_{\phi}(\mathbf{z}_j|\mathbf{z}_i) = \mathcal{N}\left(\mathbf{z}_j; \boldsymbol{\mu}_j + \boldsymbol{\gamma}_{ij} \odot \boldsymbol{\sigma}_j \odot \boldsymbol{\sigma}_i^{-1} \odot (\mathbf{z}_i - \boldsymbol{\mu}_i), \boldsymbol{\sigma}_j \odot \sqrt{\mathbf{1}_D - \boldsymbol{\gamma}_{ij}^2}\right). \qquad (7)$$

With the conditional distribution $q_{\phi}(\mathbf{z}_j|\mathbf{z}_i)$, joint samples $(\mathbf{z}_1, \mathbf{z}_2, \cdots, \mathbf{z}_N)$ can be drawn from $q_{\phi}^{\mathbb{T}}(\mathbf{Z}|\mathbf{X})$ with ancestral sampling. By sampling $\mathbf{z}_i$ one by one with ancestral sampling, we can show that the joint sample $(\mathbf{z}_1, \mathbf{z}_2, \cdots, \mathbf{z}_N)$ can be equivalently represented by a set of $N$ independent Gaussian noises $\{\boldsymbol{\epsilon}_i\}_{i=1}^N$ with $\boldsymbol{\epsilon}_i \sim \mathcal{N}(\mathbf{0}_D, \mathbf{I}_D)$, as stated in the theorem below.

**Theorem 1.** *Suppose that $N$ latent variables $\mathbf{Z} = [\mathbf{z}_1, \cdots, \mathbf{z}_N]^{\top}$ follow a tree-structured posterior distribution $q_{\phi}^{\mathbb{T}}(\mathbf{Z}|\mathbf{X})$ with the tree structure $\mathbb{T} = (\mathcal{V}, \mathcal{E})$, the joint sample $(\mathbf{z}_1, \mathbf{z}_2, \cdots, \mathbf{z}_N) \sim q_{\phi}^{\mathbb{T}}(\mathbf{Z}|\mathbf{X})$ can be expressed as*

$$\mathbf{z}_j = \boldsymbol{\mu}_j + \tilde{\boldsymbol{\gamma}}_{1j} \odot \boldsymbol{\epsilon}_1 \odot \boldsymbol{\sigma}_j + \sum_{i \in \mathbb{P}_{1 \rightarrow j}, i \neq 1} \tilde{\boldsymbol{\gamma}}_{ij} \odot \sqrt{\mathbf{1}_D - \boldsymbol{\gamma}_{\mathrm{pa}(i),i}^2} \odot \boldsymbol{\epsilon}_i \odot \boldsymbol{\sigma}_j, \quad \text{for } j \in \mathcal{V}, \qquad (8)$$

*where $\boldsymbol{\epsilon}_i \sim \mathcal{N}(\mathbf{0}_D, \mathbf{I}_D)$ is a Gaussian random noise and $\mathrm{pa}(i)$ denotes the parent node of $\mathbf{z}_i$ with respect to $i \in \mathcal{V}$.*

For the proof and a concrete example, we refer to Appendix B. With the reparameterization of sample $\mathbf{z}_j$ in Eq. (8), the joint sample $\mathbf{Z} = [\mathbf{z}_1^{\top}, \cdots, \mathbf{z}_N^{\top}]^{\top}$ can be re-parameterized in a matrix-form as

$$\mathbf{Z}^{(\mathbb{T})} = \boldsymbol{\mu}_{\mathbf{z}} + \mathbf{L}_{\mathbf{z}}^{(\mathbb{T})}\boldsymbol{\epsilon} \quad \text{with} \quad \boldsymbol{\epsilon} = [\boldsymbol{\epsilon}_1^{\top}, \cdots, \boldsymbol{\epsilon}_N^{\top}]^{\top}, \qquad (9)$$

where $\boldsymbol{\epsilon}_i \sim \mathcal{N}(\mathbf{0}_D, \mathbf{I}_D)$ for $i \in \mathcal{V}$; and $\mathbf{L}_{\mathbf{z}}^{(\mathbb{T})}$ is a $ND \times ND$ matrix, with its $(i, j)$-th block $[\mathbf{L}_{\mathbf{z}}^{(\mathbb{T})}]_{ij} = \mathrm{diag}(\boldsymbol{\ell}_{ij})$, and

$$\boldsymbol{\ell}_{ij} = \begin{cases} \tilde{\boldsymbol{\gamma}}_{1j} \odot \boldsymbol{\sigma}_j, & i = 1, j \in \mathcal{V}, \\ \tilde{\boldsymbol{\gamma}}_{ij} \odot \sqrt{\mathbf{1}_D - \boldsymbol{\gamma}_{\mathrm{pa}(i),i}^2} \odot \boldsymbol{\sigma}_j, & i \neq 1, i \in \mathbb{P}_{1 \rightarrow j}, j \in \mathcal{V}, \\ \mathbf{0}_D, & \text{otherwise}. \end{cases} \qquad (10)$$

Actually, if the variables are indexed according to their positions on the tree $\mathbb{T}$ from left to right and then top to bottom, it can be easily shown that $\mathbf{L}_{\mathbf{z}}^{(\mathbb{T})}$ is a lower-triangular block matrix. For example, the matrix $\mathbf{L}_{\mathbf{z}}^{(\mathbb{T})}$ corresponding to the quin-variate example of Fig. 1b can be written as follows

$$\mathbf{L}_{\mathbf{z}}^{(\mathbb{T})} = \mathrm{diag}(\boldsymbol{\sigma}_{\mathbf{z}}) \begin{bmatrix} \mathbf{1}_D & & & & \\ \tilde{\boldsymbol{\gamma}}_{12} & \sqrt{\mathbf{1}_D - \boldsymbol{\gamma}_{12}^2} & & & \\ \tilde{\boldsymbol{\gamma}}_{13} & \tilde{\boldsymbol{\gamma}}_{23} \odot \sqrt{\mathbf{1}_D - \boldsymbol{\gamma}_{12}^2} & \sqrt{\mathbf{1}_D - \boldsymbol{\gamma}_{23}^2} & & \\ \tilde{\boldsymbol{\gamma}}_{14} & \tilde{\boldsymbol{\gamma}}_{24} \odot \sqrt{\mathbf{1}_D - \boldsymbol{\gamma}_{12}^2} & \mathbf{0}_D & \sqrt{\mathbf{1}_D - \boldsymbol{\gamma}_{24}^2} & \\ \tilde{\boldsymbol{\gamma}}_{15} & \tilde{\boldsymbol{\gamma}}_{25} \odot \sqrt{\mathbf{1}_D - \boldsymbol{\gamma}_{12}^2} & \tilde{\boldsymbol{\gamma}}_{35} \odot \sqrt{\mathbf{1}_D - \boldsymbol{\gamma}_{23}^2} & \mathbf{0}_D & \sqrt{\mathbf{1}_D - \boldsymbol{\gamma}_{35}^2} \end{bmatrix}$$
$$(11)$$

where $\mathrm{diag}(\cdot)$ has been omitted for conciseness.

By parameterizing the mean $\boldsymbol{\mu}_i$, standard deviation $\boldsymbol{\sigma}_i$ and the correlation parameters $\boldsymbol{\gamma}_{ij}$ for $(i, j) \in \mathcal{E}$ with a neural network $f_{\phi}(\cdot, \cdot)$ as

$$\boldsymbol{\gamma}_{ij} = f_{\phi}(\mathbf{x}_i, \mathbf{x}_j), \quad (i, j) \in \mathcal{E}, \qquad (12)$$

according to Eq. (9), samples drawn from $q_{\phi}^{\mathbb{T}}(\mathbf{Z}|\mathbf{X})$ can be reparameterized in the form of neural network parameters $\phi$ and random Gaussian noise $\boldsymbol{\epsilon}$, which can facilitate the training of variational inference significantly. Importantly, except the mean and standard deviation, we only need to re-parameterize the parameters $\boldsymbol{\Gamma}^{\mathbb{T}} = \{\boldsymbol{\gamma}_{ij} : (i, j) \in \mathcal{E}\}$, while computing the other non-zero elements in $\mathbf{L}_{\mathbf{z}}^{(\mathbb{T})}$ with $\boldsymbol{\Gamma}^{\mathbb{T}}$. In this way, we only need to re-parameterize $|\mathcal{E}|$ parameters, and thus only need to run $|\mathcal{E}|$ times of neural network $f_{\phi}(\cdot, \cdot)$, instead of $\mathcal{O}(N^2)$ times in the vanilla method. For a tree, the number of edges $|\mathcal{E}| \leq N - 1$, thus we only need to additionally run $\mathcal{O}(N)$ times of neural networks $f_{\phi}(\cdot, \cdot)$, in addition to the runs required by the mean-field method. For a mean-field variational inference that assumes independence among instances, for each epoch, it also needs to run $\mathcal{O}(N)$

times of neural network. Thus, the complexity of our proposed method is roughly only 2 times of the mean-field method. As seen in the experiments on constrained clustering, the consumed time of our TreeVI is only 2 to 3 times that of mean-field variational inference methods.

It should be pointed that if we simply apply Cholesky decomposition to the correlation matrix $\mathbf{R}$ to produce $\mathbf{R} = \mathbf{L}\mathbf{L}^\top$ and then directly re-parameterize the elements $\ell_{ij}$ in the lower-triangular matrix $\mathbf{L}$ by a neural network $f_{\boldsymbol{\phi}}(\cdot, \cdot)$, that is, $\ell_{ij} = f_{\boldsymbol{\phi}}(\mathbf{x}_i, \mathbf{x}_j)$, then we have to re-parameterize as many as $N(N+1)/2$ elements, i.e., all elements from the lower-triangular positions of $\mathbf{L}$ need to be re-parameterized, where $N$ is the number of instances in training dataset. That means we need to run the neural network $f_{\boldsymbol{\phi}}(\cdot, \cdot)$ by $\mathcal{O}(N^2)$ times for every epoch, which is computationally unacceptable, especially considering that $N$ could be as large as millions in practice. And our main contribution lies at finding a way to reduce the required times of running the neural network $f_{\boldsymbol{\phi}}(\cdot, \cdot)$ from $\mathcal{O}(N^2)$ to $\mathcal{O}(N)$ by restricting the correlation matrix $\mathbf{R}$ to a special form $\mathbf{R}^{(\mathbb{T})}$ constructed from a tree $\mathbb{T} = (\mathcal{V}, \mathcal{E})$, which is actually a dense matrix with its $(i, j)$-th element $[\mathbf{R}^{(\mathbb{T})}]_{ij} = \tilde{\gamma}_{ij}$ defined as Eq. (3) for $(i, j) \notin \mathcal{E}$. Under the restricted correlation matrix $\mathbf{R}^{(\mathbb{T})}$, the lower-triangular matrix $\mathbf{L}_{\mathbf{z}}^{(\mathbb{T})}$ possesses a very elegant form, as shown in Eq. (10). The elegance lies at that although $\mathbf{L}_{\mathbf{z}}^{(\mathbb{T})}$ still has $\mathcal{O}(N^2)$ non-zero elements, all of these non-zero elements can be explicitly computed from the $|\mathcal{E}|$ parameters $\boldsymbol{\Gamma}^{\mathbb{T}} = \{\boldsymbol{\gamma}_{ij} : (i, j) \in \mathcal{E}\}$.

With the tree-structured posterior, the data log-likelihood has the evidence lower bound $\log p(\mathbf{X}) \geq \mathbb{E}_{\mathbf{Z} \sim q_{\boldsymbol{\phi}}^{\mathbb{T}}(\mathbf{Z}|\mathbf{X})}[\log p_{\boldsymbol{\theta}}(\mathbf{X}, \mathbf{Z})] + \mathcal{H}[q_{\boldsymbol{\phi}}^{\mathbb{T}}(\mathbf{Z}|\mathbf{X})]$, where the first term can be estimated by reparameterization $\mathbf{Z}^{(\mathbb{T})} \sim q_{\boldsymbol{\phi}}^{\mathbb{T}}(\mathbf{Z}|\mathbf{X})$, and the entropy $\mathcal{H}$ of the tree-structured posterior distribution $q_{\boldsymbol{\phi}}^{\mathbb{T}}(\mathbf{Z}|\mathbf{X}) = \prod_{i \in \mathcal{V}} q_{\boldsymbol{\phi}}(\mathbf{z}_i|\mathbf{x}_i) \prod_{(i,j) \in \mathcal{E}} \frac{q_{\boldsymbol{\phi}}(\mathbf{z}_i, \mathbf{z}_j|\mathbf{x}_i, \mathbf{x}_j)}{q_{\boldsymbol{\phi}}(\mathbf{z}_i|\mathbf{x}_i) q_{\boldsymbol{\phi}}(\mathbf{z}_j|\mathbf{x}_j)}$ can be decomposed as entropy terms with respect to singleton posterior $q_{\boldsymbol{\phi}}(\mathbf{z}_i|\mathbf{x}_i)$ and pairwise posterior $q_{\boldsymbol{\phi}}(\mathbf{z}_i, \mathbf{z}_j|\mathbf{x}_i, \mathbf{x}_j)$ which can be both directly computed (for detailed expressions we refer to Appendix). Therefore, the optimization of variational inference with tree-structured posterior approximation can be simplified as maximizing the following sampling-based evidence lower bound of our proposed TreeVI

$$\mathcal{L}^{\mathbb{T}}(\boldsymbol{\theta}, \boldsymbol{\phi}, \mathbf{X}) = \log p_{\boldsymbol{\theta}}(\mathbf{X}, \mathbf{Z}^{(\mathbb{T})}) + \mathcal{H}[q_{\boldsymbol{\phi}}^{\mathbb{T}}(\mathbf{Z}|\mathbf{X})], \tag{13}$$

where $\mathbf{Z}^{(\mathbb{T})}$ denotes the tree-structured reparameterization for latent variables with respect to the tree structure $\mathbb{T}$. For detailed computation for the evidence lower bound, we refer to Appendix C.1.

## 2.2 Extension to Multiple Trees

The expressiveness of a single tree-structured posterior is still restrictive. To alleviate this issue, we propose to approximate the true posterior with a mixture-of-trees posterior distribution

$$q_{\boldsymbol{\phi}}^{\mathbb{MT}}(\mathbf{Z}|\mathbf{X}) = \sum_{m=1}^{M} \pi_m q_{\boldsymbol{\phi}}^{\mathbb{T}_m}(\mathbf{Z}|\mathbf{X}), \tag{14}$$

where we use a weighted mixture of tree structures $\mathbb{MT} = \{\mathbb{T}_1, \cdots, \mathbb{T}_M\}$, as shown in Fig. 1c, to approximate the underlying correlation structure, and the posterior $q_{\boldsymbol{\phi}}^{\mathbb{T}_m}(\mathbf{Z}|\mathbf{X})$ with respect to the $m$-th tree component is expressed in the form of Eq. (4). Each tree component $\mathbb{T}_m = (\mathcal{V}, \mathcal{E}_m)$ corresponds to a set of latent indices $\mathcal{V} = \{1, \cdots, N\}$, a set of pairwise connections $\mathcal{E}_m$, a set of correlation parameters $\boldsymbol{\Gamma}^{\mathbb{T}_m} = \{\boldsymbol{\gamma}_{ij}^m : (i, j) \in \mathcal{E}_m\}$ to be learned, and a weight controlled by a tree coefficient $\pi_m$, where $m = 1, \cdots, M$.

With the mixture-of-trees posterior, the data log-likelihood has the evidence lower bound $\log p(\mathbf{X}) \geq \sum_{m=1}^{M} \pi_m \mathbb{E}_{\mathbf{Z} \sim q_{\boldsymbol{\phi}}^{\mathbb{T}_m}(\mathbf{Z}|\mathbf{X})}[\log p_{\boldsymbol{\theta}}(\mathbf{X}, \mathbf{Z})] + \mathcal{H}[q_{\boldsymbol{\phi}}^{\mathbb{MT}}(\mathbf{Z}|\mathbf{X})]$, where the first term can be estimated by tree-structured reparameterization $\mathbf{Z}^{(\mathbb{T}_m)} \sim q_{\boldsymbol{\phi}}^{\mathbb{T}_m}(\mathbf{Z}|\mathbf{X})$ within each tree component $\mathbb{T}_m$ for $m = 1, \cdots, M$, but the entropy term has no explicit expression. Since the entropy $\mathcal{H}[p]$ is concave in the probability distribution $p$, the entropy of the mixture-of-trees posterior distribution is lower bounded by the weighted sum of entropies with respect to each tree component $\sum_{m=1}^{M} \pi_m \mathcal{H}[q_{\boldsymbol{\phi}}^{\mathbb{T}_m}(\mathbf{Z}|\mathbf{X})]$, which can be similarly computed by the singular and pairwise marginal posterior distributions with respect to each tree component $\mathbb{T}_m$, $m = 1, \cdots, M$. Therefore, the optimization of variational

inference with mixture-of-trees posterior approximation can be simplified as maximizing the following sampling-based evidence lower bound of our proposed MTreeVI

$$\mathcal{L}^{\mathrm{MT}}(\boldsymbol{\theta}, \boldsymbol{\phi}, \mathbf{X}) = \sum_{m=1}^{M} \pi_m \left[ \log p_{\boldsymbol{\theta}}(\mathbf{X}, \mathbf{Z}^{(\mathbb{T}_m)}) + \mathcal{H}[q_{\boldsymbol{\phi}}^{\mathbb{T}_m}(\mathbf{Z}|\mathbf{X})] \right], \tag{15}$$

where $\mathbf{Z}^{(\mathbb{T}_m)}$ denotes the tree-structured reparameterization for latent variables with respect to the tree structure $\mathbb{T}_m$. For detailed computation for the evidence lower bound, we refer to Appendix C.2.

## 2.3 Learning the Tree Structure from Data

Our proposed tree-structured and mixture-of-trees structured posterior approximation make variational inference with pairwise latent correlations feasible, but a specific tree structure over latent variables need to be determined in advance. In this section, our goal is to develop an efficient algorithm for learning correlation-rich tree structures to approximate the underlying posterior correlation structure.

To learn a meaningful tree structure from the latent embeddings, we adopt a symmetric binary matrix $\mathbf{A} \in \{0, 1\}^{N \times N}$ with entries 0 along the diagonal, where each element $a_{ij} \in \{0, 1\}$ represents an edge between latents $\mathbf{z}_i$ and $\mathbf{z}_j$, $i \neq j \in \{1, \cdots, N\}$ (here we assume $D = 1$ for convenience).

**Proposition 1.** *Suppose that the symmetric adjacency matrix $\mathbf{A} \in \{0, 1\}^{N \times N}$, then the undirected structure induced by adjacency matrix $\mathbf{A}$ is acyclic if and only if*

$$h(\mathbf{A}) = \mathrm{tr}\, \mathbf{A} \exp(\mathbf{A}^2) = 0 \tag{16}$$

*where $\mathrm{tr}(\cdot)$ and $\exp(\cdot)$ represent the trace and exponential of a matrix respectively, and $\odot$ is the Hadamard product.*

For the proof we refer to the Appendix D.1. Inherited from the idea of Zheng et al. [45], we use a similar indicator function $h(\mathbf{A})$ in Eq. (16) to check the acyclicity of the structure induced by the symmetric binary matrix $\mathbf{A}$. If and only if $h(\mathbf{A}) = 0$, the adjacency matrix $\mathbf{A}$ determines a unique acyclic undirected graph, which can be leveraged to build a tree structure $\mathbb{T}(\mathbf{A}) = (\mathcal{V}, \mathcal{E})$ with nodes $\mathcal{V} \subset \{1, \cdots, N\}$ and edges $\mathcal{E} = \{(i, j) : a_{ij} \neq 0, i \neq j \in \mathcal{V}\}$. Under the constraints Eq. (16), we seek to establish the following continuous optimization problem

$$\min_{\boldsymbol{\theta}, \boldsymbol{\phi}, \mathbf{A}} \ell(\boldsymbol{\theta}, \boldsymbol{\phi}, \mathbf{X}, \mathbf{A}), \quad \text{subject to } h(\mathbf{A}) = 0, \tag{17}$$

where $\ell(\boldsymbol{\theta}, \boldsymbol{\phi}, \mathbf{X}, \mathbf{A}) = -\mathcal{L}^{\mathbb{T}(\mathbf{A})}(\boldsymbol{\theta}, \boldsymbol{\phi}, \mathbf{X})$ is the negative of the evidence lower bound given by Eq. (13), and the correlation parameters $\boldsymbol{\Gamma}^{\mathbb{T}(\mathbf{A})} = \{\boldsymbol{\gamma}_{ij} : a_{ij} \neq 0, i \neq j \in \mathcal{V}\}$ of the learned tree is determined by the binary adjacency matrix $\mathbf{A}$ with $\boldsymbol{\gamma}_{ij} = f_{\boldsymbol{\phi}}(\mathbf{x}_i, \mathbf{x}_j)$ calculated by the neural network. A similar optimization can be implemented to stochastically learn a meaningful mixture of trees by using a set of symmetric adjacency matrices $\mathcal{A} = \{\mathbf{A}_1, \cdots, \mathbf{A}_M\}$ to represent multiple tree components, and further maximizing the evidence lower bound Eq. (15) under acyclic constraints Eq. (16) of each matrix in $\mathcal{A}$. The constrained optimization problem above can be further converted into unconstrained subproblems with Lagrangian multiplies and efficiently solved by numerical algorithms or stochastic gradient methods. For implementation details, we refer to Appendix D.3.

To initialize the tree structure before constrained optimization, the easiest way is to randomly build from the fully connected graph over data instances by using depth-first search (DFS) algorithm. To enrich the initialized tree structure with neighboring correlation information, the uniform sampling process in the DFS algorithm can be further modified to generate a meaningful neighborhood for each instance, by assigning the probability of sampling a neighbor of each instance according to their similarity. For detailed implementation of the tree initialization, we refer to the Appendix D.2.

## 3 Related Work

Variational inference is broadly used for approximating intractable posterior in latent variable models, but notorious for its restricted variational families, especially mean-field variational family which is still widely used by modern generative models [4, 37]. So far there has been a wide variety of variational inference methods that attempt to improve on traditional mean-field variational inference by modeling dependence structures within latent posterior distribution, such as constructing the

variational distribution with a normalizing flow [30, 6, 39], which deterministically transforms a simple probability distribution over latent variable to a complex one through a sequence of invertible and differentiable functions with tractable Jacobians. Besides deterministic transformations, the form of structured variational inference can be diverse, such as constructing with implicit models [12, 35, 26], modeling dependencies between local and global parameters [11, 38], constructing with a mixture of variational distributions [16, 27, 20], determining pairwise dependencies between univariate marginals with copula functions [34, 13, 33], and designing variational distribution with hierarchies [38, 1, 25]. However, these work are mostly focused on distributional assumption over latent dimensions, that fail to be directly extended to capture instance-level correlation structure.

Recently, there has been some work on incorporating instance-level correlation structure in variational inference [24, 36, 29]. Manduchi et al. [24] designs a prior information matrix to express must-link and cannot-link constraints between data in an explicit way, and integrates the instance-level prior information into the framework of variational inference by conditioning on the prior clustering preferences. But the instance-level correlation is only considered in the generative process regardless of the correlation structure induced from latent posterior by still adopting an amortized mean-field variational distribution. The work of Tang et al. [36] is the most related to ours, which takes instance-level correlation structure into consideration when learning latent representations, but restricts both the prior and posterior distributions to be identically tree-structured for tractable optimization. However, this assumption is unrealistic in most variational inference scenarios, and high-order correlations are unable to be modeled within the tree structure.

## 4 Experiments

**Tasks & Datasets.** We evaluate our methods with four different tasks: synthetic evidence lower bound test, constrained clustering, user matching and link prediction, on synthetic dataset, standard datasets (MNIST, Fashion MNIST, Reuters and STL-10), public movie rating and product rating datasets, respectively. We refer to Appendix E.1 for more dataset details.

**Baselines & Implementation Details.** For baselines on the user matching and link prediction tasks, we include the standard variational auto-encoders [18] and a recent modification to VAE [36] taking instance-level correlation structure into consideration. And we also compare our method to the state-of-the-art method learning latent embeddings with graph convolutional networks, GraphSAGE [8]. With regard to the constrained clustering task, we take a variety of constrained clustering methods, e.g., the traditional pairwise constrained K-means (PCKmeans, [2]), deterministic deep constrained clustering method based on DEC (SDEC, [32]) and constrained IDEC (C-IDEC, [44]), as state-of-the-art constrained clustering methods for comparison. We also include the generative models, the unsupervised VaDE [14], the graph augmented VaDE (DGG, [42]), and the weakly-supervised DC-GMM [24]. For comparison, we experiment on our methods TreeVI and MTreeVI, where the number of tree components for the mixture-of-trees posterior is fixed as $M = 3$. Aadditional information related to experimental implementation details are in Appendix E.2.

### 4.1 Synthetic VAE

We design a synthetic dataset with a graph-structured latent variable model. The dataset contains $N = 6000$ data points $\mathbf{x}_1, \cdots, \mathbf{x}_N \in \mathbb{R}^D$ with $D = 4$, each independently generated from the conditional distribution $p(\mathbf{x}|\mathbf{z}) = \mathcal{N}(\mathbf{x}; \mathbf{z}, \sigma^2 \mathbf{I}_4)$ given latent embeddings $\mathbf{z}_1, \cdots, \mathbf{z}_N \in \mathbb{R}^D$ where $\sigma^2$ is a fixed value and set to 0.5. The latent embeddings $\mathbf{z}_1, \cdots, \mathbf{z}_N$ are drawn from a zero-mean normal distribution $p(\mathbf{z}) = \mathcal{N}(\mathbf{z}; \mathbf{0}_4, \mathbf{\Sigma_z})$, and the graph-structured correlation is incorporated into the latent covariance matrix $\mathbf{\Sigma_z} = \mathbf{I}_4 + \lambda \mathbf{A}$ via an affinity matrix $\mathbf{A} \in \mathbb{R}^{4 \times 4}$ assigned with a loopy graph structure

$$\mathbf{A} = \begin{bmatrix} 0 & 1 & 0 & 0.3 \\ 1 & 0 & 1 & 0.3 \\ 0 & 1 & 0 & 0.4 \\ 0.3 & 0.3 & 0.4 & 0 \end{bmatrix} \quad (18)$$

Table 1: Estimated lower bounds (ELBO) of VAE with posterior distributions approximation by mean-field distributions, tree structures $\mathbb{T}_1$ and $\mathbb{T}_2$, as well as their mixture model MTreeVI($\mathbb{T}_1$,$\mathbb{T}_2$), compared to ground truth log-likelihood $\log p(\mathbf{X})$.

| Methods | Lower Bound |
|---|---|
| Mean-field | -11.1535 |
| TreeVI ($\mathbb{T}_1$) | -10.8998 |
| TreeVI ($\mathbb{T}_2$) | -10.6137 |
| MTreeVI | -10.3586 |
| $\log p(\mathbf{X})$ | -10.3417 |

Table 2: Clustering performances (%) of our proposed methods TreeVI and MTreeVI compared with baselines. Means and standard deviations are computed across 10 runs with different random initializations. † Results taken from DC-GMM [24]

| Dataset | Metric | VaDE† | SDEC† | C-IDEC† | DGG | DC-GMM | TreeVI | MTreeVI |
|---|---|---|---|---|---|---|---|---|
| MNIST | ACC | 89.0 ±5.0 | 86.2 ±0.1 | 96.3 ±0.2 | 95.8 ±0.1 | 96.5 ±0.2 | **97.4 ±0.3** | **97.5 ±0.4** |
| | NMI | 82.8 ±3.0 | 84.2 ±0.1 | 91.8 ±1.0 | 91.2 ±0.2 | 91.4 ±0.3 | **93.1 ±0.6** | **93.1 ±0.6** |
| | ARI | 80.9 ±5.0 | 80.1 ±0.1 | 92.1 ±0.4 | 91.4 ±0.3 | 92.5 ±0.5 | **93.7 ±0.7** | **94.0 ±0.5** |
| fMNIST | ACC | 55.1 ±2.2 | 54.0 ±0.2 | 68.1 ±3.0 | 79.9 ±0.4 | 80.5 ±0.8 | **81.4 ±0.6** | **82.1 ±0.7** |
| | NMI | 57.9 ±2.7 | 57.3 ±0.1 | 66.7 ±2.0 | 70.1 ±0.3 | 72.0 ±0.4 | **73.9 ±0.6** | **74.1 ±0.6** |
| | ARI | 41.6 ±3.1 | 40.2 ±0.1 | 52.3 ±3.0 | 64.9 ±0.3 | 66.4 ±0.5 | **67.9 ±0.9** | **68.1 ±0.6** |
| Reuters | ACC | 76.0 ±0.7 | 82.1 ±0.1 | 94.7 ±0.6 | 93.5 ±0.6 | 95.4 ±0.2 | **95.9 ±0.6** | **96.1 ±0.7** |
| | NMI | 50.1 ±1.3 | 62.3 ±0.1 | 81.4 ±0.7 | 81.2 ±0.8 | 82.7 ±0.7 | **83.4 ±0.5** | **83.9 ±0.5** |
| | ARI | 58.0 ±1.4 | 66.7 ±0.1 | 87.7 ±0.9 | 87.8 ±0.5 | 89.0 ±0.6 | **90.2 ±0.4** | **90.5 ±0.4** |
| STL-10 | ACC | 77.3 ±0.5 | 79.2 ±0.1 | 81.6 ±3.8 | 89.9 ±0.3 | 89.5 ±0.5 | **90.4 ±0.9** | **90.7 ±0.9** |
| | NMI | 70.6 ±0.4 | 78.6 ±0.1 | 77.3 ±1.7 | 80.9 ±0.5 | 80.2 ±0.7 | **81.3 ±0.8** | **81.6 ±0.7** |
| | ARI | 62.7 ±0.4 | 71.0 ±0.1 | 71.8 ±3.4 | 79.0 ±0.4 | 78.4 ±0.9 | **79.5 ±0.7** | **79.7 ±0.9** |

where $\lambda$ is leveraged to control the overall correlation strength and set to 0.5. Two tree structures over latent dimensions $\mathbb{T}_1 = (\mathcal{V}, \mathcal{E}_1)$ and $\mathbb{T}_2 = (\mathcal{V}, \mathcal{E}_2)$ are designed with vertex set $\mathcal{V} = \{1, 2, 3, 4\}$ and edge sets $\mathcal{E}_1 = \{(1, 2), (1, 3), (2, 4)\}$ and $\mathcal{E}_2 = \{(1, 2), (1, 4), (2, 3)\}$, respectively. The posterior correlation structure is modeled by $\mathbb{T}_1$, $\mathbb{T}_2$ and their mixture model MTreeVI($\mathbb{T}_1, \mathbb{T}_2$), with the estimated evidence lower bounds and ground truth log-likelihood shown in Table 1. It can be seen that tree-structured posteriors can learn more correlation information than traditional mean-field approximations. Moreover, different choices of the tree structures influence the amount of correlation information, and mixture of tree components can obtain more correlations than each individual.

## 4.2 Constrained Clustering

Constrained clustering tasks differ from the classic clustering scenario with access to instance-level constraints, consisting of *must-links* if two samples are believed to belong to the same cluster, and *cannot-links*, otherwise. Based on the variational deep embedding (VaDE) framework [14], constrained clustering can be formulated as a probabilistic clustering problem with joint probability $p_{\boldsymbol{\theta}}(\mathbf{X}, \mathbf{Z}, \mathbf{c}) = p_{\boldsymbol{\theta}}(\mathbf{X}|\mathbf{Z})p(\mathbf{Z}|\mathbf{c})p(\mathbf{c})$, where the sample $\mathbf{x}_i$ is generated from a normal distribution conditioned on $\mathbf{z}_i$, $\mathbf{z}_i$ is sampled from $p(\mathbf{z}_i|c_i) = \mathcal{N}(\mathbf{z}_i; \boldsymbol{\mu}_{c_i}, \mathrm{diag}(\boldsymbol{\sigma}_{c_i}^2))$, and the cluster assignments $\mathbf{c} = \{c_i\}_{i=1}^N$ are sampled from a categorical distribution. Following previous work [24], we incorporate the clustering preference through a conditional probability $p(\mathbf{c}|\mathbf{W})$ with a pairwise prior information matrix $\mathbf{W}$

$$p(\mathbf{c}|\mathbf{W}) := \frac{\prod_i \pi_{c_i} h_i(\mathbf{c}, \mathbf{W})}{\sum_{\mathbf{c}} \prod_j \pi_{c_j} h_j(\mathbf{c}, \mathbf{W})} = \frac{1}{\Omega(\boldsymbol{\pi})} \prod_i \pi_{c_i} h_i(\mathbf{c}, \mathbf{W}) \tag{19}$$

where $\boldsymbol{\pi} = \{\pi_k\}_{k=1}^K$ are the weights associated to each cluster, $\Omega(\boldsymbol{\pi})$ is the normalization factor and $h_i(\mathbf{c}, \mathbf{W}) = \prod_{j \neq i} \exp(\mathbf{W}_{ij}\delta_{c_i c_j})$ is a weighting function. The pairwise prior information matrix $\mathbf{W}$ is defined as a symmetric matrix containing the pairwise constraints: $\mathbf{W}_{ij} > 0$ if there is a must-link constraint between $\mathbf{x}_i$ and $\mathbf{x}_j$, $\mathbf{W}_{ij} < 0$ if there is a cannot-link constraint between $\mathbf{x}_i$ and $\mathbf{x}_j$, and $\mathbf{W}_{ij} = 0$ otherwise. The values $|\mathbf{W}_{ij}| \in [0, \infty)$ reflect the degree of certainty in the constraints, and are set to $10^4$ for all datasets. And 6000 pairwise constraints are used for experiments on both our methods and other constrained clustering baselines.

The variational posterior distribution is defined as $q_{\boldsymbol{\phi}}(\mathbf{Z}, \mathbf{c}|\mathbf{X}) = q_{\boldsymbol{\phi}}(\mathbf{Z}|\mathbf{X})q(\mathbf{c}|\mathbf{Z})$, where the probability of cluster assignments is factorized as $q(\mathbf{c}|\mathbf{Z}) = \prod_i q(c_i|\mathbf{z}_i)$ which can be easily computed by Bayes theorem. In the work of DC-GMM [24], the posterior distribution $q_{\boldsymbol{\phi}}(\mathbf{Z}|\mathbf{X})$ for latent variables is assumed to be mean-field which fails to capture the posterior correlation structure, while in our methods TreeVI and MTreeVI, the latent posterior is approximated by tree-structured and mixture-of-trees distribution, respectively. In Table 2 we report the averaged clustering performances across 10 turns of both our proposed methods against baseline methods. Accuracy (ACC), Normalized Mutual Information (NMI), and Adjusted Rand Index (ARI) are used as evaluation metrics. It can be

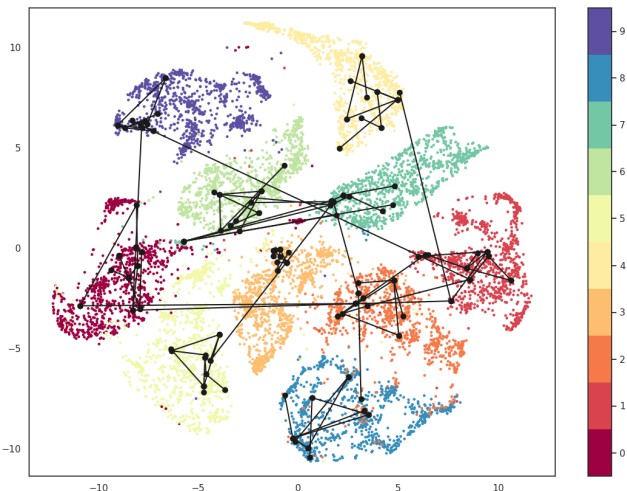

Figure 2: T-SNE visualization of MNIST samples in the embedded space and the learnt tree structure of our proposed TreeVI. 100 samples are randomly selected to plot their instance-level tree structure (colored in **black**).

easily observed that our models reach the state-of-the-art clustering performance in all metrics and datasets, benefiting from the correlation structures captured by our designed correlated posteriors. Moreover, due to high efficiency of tree-structured reparameterization, our methods have computational complexity comparable to DC-GMM that adopts the fully factorized posterior distribution. For example, the training of DC-GMM on MNIST dataset each epoch takes 2 to 3 seconds on GeForce RTX 3090, while our TreeVI and MTreeVI take around 5 and 9 seconds, respectively. Further, we plot the tree learned over the constrained clustering experiment on MNIST dataset, as shown in Figure 2. From the figure, we can see that instances from the same category are connected more tightly than those from different categories in the learned tree. This demonstrates that our method has the abilities to capture the underlying inherent correlations among different instances.

### 4.3 User Matching

We evaluate our methods against the baselines on a public movie rating dataset **MovieLens 20M**. The ratings of each user $u_i$ are binarized as a bag-of-word vector $\mathbf{x}_{u_i}$ ($i = 1, 2, \cdots, N$), and only ratings for movies that have been rated over 1000 times are preserved for simplicity. In our experiments, the watch history of each user $u_i$ is randomly split into two halves, leading to two synthetic users $u_i^A$ and $u_i^B$ with the most similar movie preference, and a correlation graph $\mathbb{G} = (\mathcal{V}, \mathcal{E})$ with nodes $\mathcal{V} = \{u_i^A, u_i^B : i = 1, \cdots, N\}$ and edges $\mathcal{E} = \{(u_i^A, u_i^B) : i = 1, \cdots, N\}$.

Suppose that the joint distribution of rating data $\mathbf{X}$ and the corresponding latent embeddings $\mathbf{Z}$ is modeled by $p_{\boldsymbol{\theta}}(\mathbf{X}, \mathbf{Z}) = p_{\boldsymbol{\theta}}(\mathbf{X}|\mathbf{Z})p(\mathbf{Z})$, and the variational posterior distribution is modeled by $q_{\boldsymbol{\phi}}(\mathbf{Z}|\mathbf{X})$. In the work of CVAE [36], both the prior distribution $p(\mathbf{Z})$ and posterior distribution $q_{\boldsymbol{\phi}}(\mathbf{Z}|\mathbf{X})$ are designed as weighted sums of tree-structured distributions with respect to each maximal acyclic subgraph of the correlation graph $\mathbb{G}$. While in our work, the latent representations are drawn from a Gaussian distribution of the form $p(\mathbf{Z}) = \mathcal{N}(\mathbf{Z}; \mathbf{0}_{2N}; (\mathbf{I}_{2N} + \lambda\mathbf{A}) \otimes \mathbf{I}_D)$, where the hyper-parameter $\lambda \in (0, 1)$ is used to control the overall correlation strength, and the affinity matrix $\mathbf{A} = [a_{ij}]_{i,j \in \mathcal{V}}$ is derived from the correlation graph with $a_{ij} = 1$ if $(i, j) \in \mathcal{E}$ and $a_{ij} = 0$ otherwise. And our latent posterior distribution $q_{\boldsymbol{\phi}}(\mathbf{Z}|\mathbf{X})$ is modeled by tree-structured and mixture-of-trees distributions, respectively.

And our goal is to identify the dual user $u_i^B$ given a synthetic user $u_i^A$ from a held-out set in terms of the latent embedding distance. The user data are implemented a random train/test split with a 90/10 ratio, and the synthetic user pairs from the training set are used to train all the methods. To evaluate the user matching accuracies, a fixed number $N^{\text{eval}} = 1000$ of synthetic user pairs are selected from the test set and for each synthetic user $u_i^A$ (or $u_i^B$), we summarize the ranking of its dual user $u_i^B$ (or $u_i^A$) among all other $2N^{\text{eval}} - 1$ synthetic user candidates in terms of latent embedding distances.

Table 3: Synthetic user matching test RR

| Methods | Test RR |
|---|---|
| VAE | $0.3498 \pm 0.0167$ |
| $\text{CVAE}_{\text{ind}}$ | $0.6608 \pm 0.0066$ |
| $\text{CVAE}_{\text{corr}}$ | $0.7129 \pm 0.0096$ |
| TreeVI (Ours) | $\mathbf{0.7408 \pm 0.0124}$ |
| MTreeVI (Ours) | $\mathbf{0.7521 \pm 0.0101}$ |

Table 4: Link prediction test Normalized CRR

| Methods | Test NCRR |
|---|---|
| VAE | $0.0052 \pm 0.0007$ |
| GraphSAGE | $0.0115 \pm 0.0025$ |
| $\text{CVAE}_{\text{ind}}$ | $0.0160 \pm 0.0004$ |
| $\text{CVAE}_{\text{corr}}$ | $0.0171 \pm 0.0009$ |
| TreeVI (Ours) | $\mathbf{0.0188 \pm 0.0007}$ |
| MTreeVI (Ours) | $\mathbf{0.0203 \pm 0.0014}$ |

In Table 3 we show the average Reciprocal Ranking (RR) for all the methods, which demonstrates superiority of our model over both implementations for the baseline method CVAE.

### 4.4 Link Prediction

We perform link prediction task on a constructed undirected correlation graph $\mathbb{G} = (\mathcal{V}, \mathcal{E})$ within the public product rating dataset **Epinions**. The rating data are binarized into bag-of-words feature vectors $\mathbf{x}_{u_i}$ for each user $u_i$, and only products that have been rated at least 100 times are kept and users who have rated these products at least once are considered. To construct the undirected graph $\mathbb{G}$ from the single-directional "trust" statements between users $u_i, u_j \in \mathcal{V}$ provided by the dataset, we only build an edge $(u_i, u_j) \in \mathcal{E}$ if both $u_i$ trusts $u_j$ and $u_j$ trusts $u_i$. The experiment setting of our TreeVI and MTreeVI are similar to the user matching task, by defining a correlation graph incorporated prior distribution and approximating latent posterior distribution with tree-structured and mixture-of-trees distributions, respectively.

The product rating dataset is split for each user $u_i \in \mathcal{V}$ into training and test sets, with $\max\left(1, \frac{1}{20} \cdot \text{degree}(u_i)\right)$ edges held out for test edge set $\mathcal{E}_{\text{test}}$. The remaining edges for the training edge set $\mathcal{E}_{\text{train}}$ are used to train all methods on the product rating data. To evaluate the link prediction accuracies, we calculate for each user $u_i$ the Normalized Cumulative Reciprocal Rank $\text{NCRR}_i$ of the ratings of $u_i$'s test edges among all possible connections except for the training edges, in terms of latent embedding distance metrics. Formally, the NCRR value is the $[0, 1]$-normalization of the Cumulative Reciprocal Rank (CRR) formulated as $\text{CRR}_i = \sum_{(u_i, u_j) \in \mathcal{E}_{\text{test}}} |\{k : (u_i, u_k) \notin \mathcal{E}_{\text{train}}, d_{ik} \leq d_{ij}\}|^{-1}$, where $d_{ij}$ represents the latent embedding distance between users $u_i$ and $u_j$ for $1 \leq i \neq j \leq N$. Larger $\text{NCRR}_i$ indicates better ability to predict held-out test links with respect to each user $u_i$, and the averaged results for all methods are reported in Table 4.

## 5 Conclusion

In this work, we present a novel variational inference method called TreeVI, that approximates the intractable latent posterior distribution with a tree-structured distribution. This induces a Bayesian network whose ancestral sampling gives a matrix-form reparameterization for the correlated latents and enables efficient optimization. To enrich the correlation structure, TreeVI is further extended to MTreeVI with a mixture of trees to better approximate the underlying posterior. Furthermore, the tree and mixture-of-trees structures are allowed stochastically learned from data by solving a constrained optimization problem under our proposed acyclicity constraints. With correlated posteriors, we show that our methods can capture more correlation information and achieve superior performances in real-world tasks.

**Limitations & Future Work** The proposed method requires a tree structure to approximate the graph-structured posterior correlation structure. This limitation is mitigated by adopting a weighted mixture of trees and stochastically learning a correlation-rich tree or mixture-of-trees structure with our proposed constrained optimization. For future work, we will further investigate on correlation structures with higher expressivity for approximating the latent posterior.

**Acknowledgment** This work is supported by the National Natural Science Foundation of China (No. 62276280, U1811264), Guangzhou Science and Technology Planning Project (No. 2024A04J9967).

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

## A  Precision Matrix of Tree-structured Posterior

When latent variables $\mathbf{Z} = [\mathbf{z}_1, \cdots, \mathbf{z}_N]^\top$ are all Gaussian, it is a well-known consequence of the Hammersley-Clifford theorem that the entries of the precision matrix $\mathbf{P_z} = \mathbf{\Sigma_z}^{-1}$ correspond to rescaled conditional correlations. Denote $\mathbf{P_z} = [\text{diag}(\mathbf{p}_{ij})]_{i,j=1,\cdots,N}$ with $\mathbf{p}_{ij} \in \mathbb{R}^D$, then the magnitude of $\mathbf{p}_{ij}$ expresses the correlation of $\mathbf{z}_i$ and $\mathbf{z}_j$ conditioned on other latent variables. In particular, the sparsity pattern of $\mathbf{p}_{ij}$ reflects the edge structure of the correlation graph: $\mathbf{p}_{ij} = \mathbf{0}_D$ if and only if $\mathbf{z}_i \perp\!\!\!\perp \mathbf{z}_j | \mathbf{z}_{\{1,\cdots,N\}\setminus\{i,j\}}$.

The tree structure $\mathbb{T} = (\mathcal{V}, \mathcal{E})$ with non-adjacent correlation parameters defined as Eq. (3) forms a Markov random field with joint probability given by Eq. (4), which follows the principle of conditional independence. Therefore, the precision vector $\mathbf{p}_{ij} \neq \mathbf{0}_D$ if and only if latent variables $\mathbf{z}_i$ and $\mathbf{z}_j$ are adjacent, that is, $(i,j) \in \mathcal{E}$.

**Example.** In the example of tree-structured correlation approximation with 5 latent variables as shown in Fig. 1b, the precision matrix corresponding to the tree-structured latent covariance matrix can be expressed as

$$[\mathbf{\Sigma_z}^{(\mathbb{T})}]^{-1} = \text{diag}(\boldsymbol{\sigma_z}^{-1})[\mathbf{R}^{(\mathbb{T})}]^{-1} \, \text{diag}(\boldsymbol{\sigma_z}^{-1}), \tag{20}$$

with the inverse of tree-structured correlation matrix given by

$$[\mathbf{R}^{(\mathbb{T})}]^{-1} = \begin{bmatrix} \mathbf{s}_{11} & \mathbf{s}_{12} & & & \\ \mathbf{s}_{12} & \mathbf{s}_{22} & \mathbf{s}_{23} & \mathbf{s}_{24} & \\ & \mathbf{s}_{23} & \mathbf{s}_{33} & & \mathbf{s}_{35} \\ & \mathbf{s}_{24} & & \mathbf{s}_{44} & \\ & & \mathbf{s}_{35} & & \mathbf{s}_{55} \end{bmatrix}, \tag{21}$$

demonstrating that only correlation parameters along the edge set $\mathcal{E} = \{(1,2), (2,3), (2,4), (3,5)\}$ of tree structure $\mathbb{T}$ has been essentially captured in the underlying correlation structure, where all of the matrix elements are simply combinations of correlation parameters from $\mathbf{\Gamma}^{\mathbb{T}} = \{\boldsymbol{\gamma}_{12}, \boldsymbol{\gamma}_{23}, \boldsymbol{\gamma}_{24}, \boldsymbol{\gamma}_{35}\}$:

$$\mathbf{s}_{11} = (\mathbf{1}_D - \boldsymbol{\gamma}_{12}^2)^{-1}, \tag{22}$$

$$\mathbf{s}_{12} = -\boldsymbol{\gamma}_{12} \odot (\mathbf{1}_D - \boldsymbol{\gamma}_{12}^2)^{-1}, \tag{23}$$

$$\mathbf{s}_{22} = (\mathbf{1}_D + 2\boldsymbol{\gamma}_{12}^2 \odot \boldsymbol{\gamma}_{23}^2 \odot \boldsymbol{\gamma}_{24}^2 - \boldsymbol{\gamma}_{12}^2 \odot \boldsymbol{\gamma}_{23}^2 - \boldsymbol{\gamma}_{12}^2 \odot \boldsymbol{\gamma}_{24}^2 - \boldsymbol{\gamma}_{23}^2 \odot \boldsymbol{\gamma}_{24}^2) \tag{24}$$
$$\odot (\mathbf{1}_D - \boldsymbol{\gamma}_{12}^2)^{-1} \odot (\mathbf{1}_D - \boldsymbol{\gamma}_{23}^2)^{-1} \odot (\mathbf{1}_D - \boldsymbol{\gamma}_{24}^2)^{-1},$$

$$\mathbf{s}_{23} = -\boldsymbol{\gamma}_{23} \odot (\mathbf{1}_D - \boldsymbol{\gamma}_{23}^2)^{-1}, \tag{25}$$

$$\mathbf{s}_{24} = -\boldsymbol{\gamma}_{24} \odot (\mathbf{1}_D - \boldsymbol{\gamma}_{24}^2)^{-1}, \tag{26}$$

$$\mathbf{s}_{33} = (\mathbf{1}_D - \boldsymbol{\gamma}_{23}^2 \odot \boldsymbol{\gamma}_{35}^2) \odot (\mathbf{1}_D - \boldsymbol{\gamma}_{23}^2)^{-1} \odot (\mathbf{1}_D - \boldsymbol{\gamma}_{35}^2)^{-1}, \tag{27}$$

$$\mathbf{s}_{35} = -\boldsymbol{\gamma}_{35} \odot (\mathbf{1}_D - \boldsymbol{\gamma}_{35}^2)^{-1}, \tag{28}$$

$$\mathbf{s}_{44} = (\mathbf{1}_D - \boldsymbol{\gamma}_{24}^2)^{-1}, \tag{29}$$

$$\mathbf{s}_{55} = (\mathbf{1}_D - \boldsymbol{\gamma}_{35}^2)^{-1}. \tag{30}$$

## B  Ancestral Sampling for Tree-structured Posterior

### B.1  Proof of Theorem 1

Suppose that $N$ latent variables $\mathbf{Z} = [\mathbf{z}_1, \cdots, \mathbf{z}_N]^\top$ follow a tree-structured posterior distribution given the tree structure $\mathbb{T} = (\mathcal{V}, \mathcal{E})$ with $\mathcal{V} = \{1, \cdots, N\}$, the ancestral sampling for each latent variable $\mathbf{z}_j$, $j \in \mathcal{V}$ can be expressed by

$$\mathbf{z}_j = \boldsymbol{\mu}_j + \tilde{\boldsymbol{\gamma}}_{1j} \odot \boldsymbol{\epsilon}_1 \odot \boldsymbol{\sigma}_j + \sum_{i \in \mathbb{P}_{1 \to j}, i \neq 1} \tilde{\boldsymbol{\gamma}}_{ij} \odot \sqrt{\mathbf{1}_D - \boldsymbol{\gamma}_{\text{pa}(i),i}^2} \odot \boldsymbol{\epsilon}_i \odot \boldsymbol{\sigma}_j, \quad j \in \mathcal{V}, \tag{31}$$

where $\boldsymbol{\epsilon}_i \sim \mathcal{N}(\mathbf{0}_D, \mathbf{I}_D)$ is a randomly sampled noise from the standard normal distribution and $\text{pa}(i)$ denotes the parent node of $\mathbf{z}_i$ with respect to $i \in \mathcal{V}$.

*Proof.* For each latent variable $\mathbf{z}_j$, there exist a unique path $\mathbb{P}_{1 \to j} = (v_1, v_2, \cdots, v_n)$ along the tree-structured Bayesian network that starts from the root node $v_1 = 1$ and ends at the goal node $v_n = j$. The joint distribution along the path $\mathbb{P}_{1 \to j}$ is given by

$$q_\phi(\mathbf{z}_{v_1}, \mathbf{z}_{v_2}, \cdots, \mathbf{z}_{v_n}) = q_\phi(\mathbf{z}_{v_1}) \prod_{i=1}^{n} q_\phi(\mathbf{z}_{v_{i+1}} | \mathbf{z}_{v_i}). \tag{32}$$

The ancestral sampling starts by sampling from $q_\phi(\mathbf{z}_{v_1})$: $\mathbf{z}_{v_1} = \boldsymbol{\mu}_{v_1} + \boldsymbol{\sigma}_{v_1} \odot \boldsymbol{\epsilon}_{v_1}$, where $\boldsymbol{\epsilon}_{v_1} \sim \mathcal{N}(\mathbf{0}_D, \mathbf{I}_D)$. Given the observation of $\mathbf{z}_{v_1}$, then the latent variable $\mathbf{z}_{v_2}$ can be sampled using the conditional distribution Eq. (7):

$$\mathbf{z}_{v_2} = \boldsymbol{\mu}_{v_2} + \boldsymbol{\gamma}_{v_1 v_2} \odot \boldsymbol{\sigma}_{v_2} \odot \boldsymbol{\sigma}_{v_1}^{-1} \odot (\mathbf{z}_{v_1} - \boldsymbol{\mu}_{v_1}) + \sqrt{\mathbf{1}_D - \boldsymbol{\gamma}_{v_1 v_2}^2} \odot \boldsymbol{\epsilon}_{v_2} \odot \boldsymbol{\sigma}_{v_2} \tag{33}$$

$$= \boldsymbol{\mu}_{v_2} + \boldsymbol{\gamma}_{v_1 v_2} \odot \boldsymbol{\epsilon}_{v_1} \odot \boldsymbol{\sigma}_{v_2} + \sqrt{\mathbf{1}_D - \boldsymbol{\gamma}_{v_1 v_2}^2} \odot \boldsymbol{\epsilon}_{v_2} \odot \boldsymbol{\sigma}_{v_2}. \tag{34}$$

Assume that at the $k$-th step of the ancestral sampling, we have

$$\mathbf{z}_{v_k} = \boldsymbol{\mu}_{v_k} + \tilde{\boldsymbol{\gamma}}_{v_1 v_k} \odot \boldsymbol{\epsilon}_{v_1} \odot \boldsymbol{\sigma}_{v_k} + \sum_{i=2}^{k} \tilde{\boldsymbol{\gamma}}_{v_i v_k} \odot \sqrt{\mathbf{1}_D - \boldsymbol{\gamma}_{v_{i-1} v_i}^2} \odot \boldsymbol{\epsilon}_{v_i} \odot \boldsymbol{\sigma}_{v_k}, \tag{35}$$

where $\tilde{\boldsymbol{\gamma}}_{v_i v_k} = \boldsymbol{\gamma}_{v_i v_{i+1}} \odot \cdots \odot \boldsymbol{\gamma}_{v_{k-1} v_k}$ for $i < k$ and $\tilde{\boldsymbol{\gamma}}_{v_k v_k} = \mathbf{1}_D$. Then at the $(k+1)$-th step of the ancestral sampling, the latent variable $\mathbf{z}_{v_{k+1}}$ can be similarly sampled by conditional sampling

$$\mathbf{z}_{v_{k+1}} = \boldsymbol{\mu}_{v_{k+1}} + \boldsymbol{\gamma}_{v_k v_{k+1}} \odot \boldsymbol{\sigma}_{v_{k+1}} \odot \boldsymbol{\sigma}_{v_k}^{-1} \odot (\mathbf{z}_{v_k} - \boldsymbol{\mu}_{v_k}) + \sqrt{\mathbf{1}_D - \boldsymbol{\gamma}_{v_k v_{k+1}}^2} \odot \boldsymbol{\epsilon}_{v_{k+1}} \odot \boldsymbol{\sigma}_{v_{k+1}} \tag{36}$$

$$= \boldsymbol{\mu}_{v_{k+1}} + \tilde{\boldsymbol{\gamma}}_{v_1 v_{k+1}} \odot \boldsymbol{\epsilon}_{v_1} \odot \boldsymbol{\sigma}_{v_{k+1}} + \sum_{i=2}^{k+1} \tilde{\boldsymbol{\gamma}}_{v_i v_{k+1}} \odot \sqrt{\mathbf{1}_D - \boldsymbol{\gamma}_{v_{i-1} v_i}^2} \odot \boldsymbol{\epsilon}_{v_i} \odot \boldsymbol{\sigma}_{v_{k+1}} \tag{37}$$

Therefore by induction, the ancestral sampling for the latent variable $\mathbf{z}_j$ can be expressed as

$$\mathbf{z}_{v_j} = \boldsymbol{\mu}_{v_j} + \tilde{\boldsymbol{\gamma}}_{v_1 v_j} \odot \boldsymbol{\epsilon}_{v_1} \odot \boldsymbol{\sigma}_{v_j} + \sum_{i=2}^{j} \tilde{\boldsymbol{\gamma}}_{v_i v_j} \odot \sqrt{\mathbf{1}_D - \boldsymbol{\gamma}_{v_{i-1} v_i}^2} \odot \boldsymbol{\epsilon}_{v_i} \odot \boldsymbol{\sigma}_{v_j}, \tag{38}$$

which is consistent with the expression of Eq. (31). $\qquad \square$

## B.2 Example of Ancestral Sampling Procedure

For the quin-variate example in Fig. 1b, we assume $\mathbf{z}_1$ as the root node, and the detailed ancestral sampling procedure for the tree-structured Bayesian network Eq. (6) is shown as follows.

1. Start by sampling from $q_\phi(\mathbf{z}_1)$:

$$\mathbf{z}_1 = \boldsymbol{\mu}_1 + \boldsymbol{\sigma}_1 \odot \boldsymbol{\epsilon}_1, \tag{39}$$

where $\boldsymbol{\epsilon}_1 \sim \mathcal{N}(\mathbf{0}_D, \mathbf{I}_D)$;

2. Then sample from $q_\phi(\mathbf{z}_2 | \mathbf{z}_1)$ given the observation of $\mathbf{z}_1$:

$$\mathbf{z}_2 = \boldsymbol{\mu}_2 + \boldsymbol{\gamma}_{12} \odot \boldsymbol{\sigma}_2 \odot \boldsymbol{\epsilon}_1 + \sqrt{\mathbf{1}_D - \boldsymbol{\gamma}_{12}^2} \odot \boldsymbol{\sigma}_2 \odot \boldsymbol{\epsilon}_2, \tag{40}$$

where $\boldsymbol{\epsilon}_2 \sim \mathcal{N}(\mathbf{0}_D, \mathbf{I}_D)$;

3. Then sample from $q_\phi(\mathbf{z}_3|\mathbf{z}_2)$ given the observation of $\mathbf{z}_2$:

$$
\begin{aligned}
\mathbf{z}_3 &= \boldsymbol{\mu}_3 + \boldsymbol{\gamma}_{23} \odot \boldsymbol{\sigma}_3 \odot \left( \boldsymbol{\gamma}_{12} \odot \boldsymbol{\epsilon}_1 + \sqrt{\mathbf{1}_D - \boldsymbol{\gamma}_{12}^2} \odot \boldsymbol{\epsilon}_2 \right) + \sqrt{\mathbf{1}_D - \boldsymbol{\gamma}_{23}^2} \odot \boldsymbol{\sigma}_3 \odot \boldsymbol{\epsilon}_3 \\
&= \boldsymbol{\mu}_3 + \boldsymbol{\gamma}_{12} \odot \boldsymbol{\gamma}_{23} \odot \boldsymbol{\sigma}_3 \odot \boldsymbol{\epsilon}_1 + \boldsymbol{\gamma}_{23} \odot \sqrt{\mathbf{1}_D - \boldsymbol{\gamma}_{12}^2} \odot \boldsymbol{\sigma}_3 \odot \boldsymbol{\epsilon}_2 \\
&\quad + \sqrt{\mathbf{1}_D - \boldsymbol{\gamma}_{23}^2} \odot \boldsymbol{\sigma}_3 \odot \boldsymbol{\epsilon}_3
\end{aligned}
\tag{41}
$$

where $\boldsymbol{\epsilon}_3 \sim \mathcal{N}(\mathbf{0}_D, \mathbf{I}_D)$;

4. Then sample from $q_\phi(\mathbf{z}_4|\mathbf{z}_2)$ given the observation of $\mathbf{z}_2$:

$$
\begin{aligned}
\mathbf{z}_4 &= \boldsymbol{\mu}_4 + \boldsymbol{\gamma}_{12} \odot \boldsymbol{\gamma}_{24} \odot \boldsymbol{\sigma}_4 \odot \boldsymbol{\epsilon}_1 + \boldsymbol{\gamma}_{24} \odot \sqrt{\mathbf{1}_D - \boldsymbol{\gamma}_{12}^2} \odot \boldsymbol{\sigma}_4 \odot \boldsymbol{\epsilon}_2 \\
&\quad + \sqrt{\mathbf{1}_D - \boldsymbol{\gamma}_{24}^2} \odot \boldsymbol{\sigma}_4 \odot \boldsymbol{\epsilon}_4
\end{aligned}
\tag{42}
$$

where $\boldsymbol{\epsilon}_4 \sim \mathcal{N}(\mathbf{0}_D, \mathbf{I}_D)$;

5. Finally, sample from $q_\phi(\mathbf{z}_5|\mathbf{z}_3)$ given the observation of $\mathbf{z}_3$:

$$
\begin{aligned}
\mathbf{z}_5 &= \boldsymbol{\mu}_5 + \boldsymbol{\gamma}_{12} \odot \boldsymbol{\gamma}_{23} \odot \boldsymbol{\gamma}_{35} \odot \boldsymbol{\sigma}_5 \odot \boldsymbol{\epsilon}_1 + \boldsymbol{\gamma}_{23} \odot \boldsymbol{\gamma}_{35} \odot \sqrt{\mathbf{1}_D - \boldsymbol{\gamma}_{12}^2} \odot \boldsymbol{\sigma}_5 \odot \boldsymbol{\epsilon}_2 \\
&\quad + \boldsymbol{\gamma}_{35} \odot \sqrt{\mathbf{1}_D - \boldsymbol{\gamma}_{23}^2} \odot \boldsymbol{\sigma}_5 \odot \boldsymbol{\epsilon}_3 + \sqrt{\mathbf{1}_D - \boldsymbol{\gamma}_{35}^2} \odot \boldsymbol{\sigma}_5 \odot \boldsymbol{\epsilon}_5
\end{aligned}
\tag{43}
$$

where $\boldsymbol{\epsilon}_5 \sim \mathcal{N}(\mathbf{0}_D, \mathbf{I}_D)$;

It is easily noted that, the expression of each latent sample in this ancestral sampling procedure is consistent to Eq. (8).

## C   Evidence Lower Bound

### C.1   Evidence Lower Bound for TreeVI

For a tree-structured posterior approximation with respect to the tree structure $\mathbb{T} = (\mathcal{V}, \mathcal{E})$, the evidence lower bound of our proposed TreeVI is given by

$$
\mathcal{L}^{\mathbb{T}}(\boldsymbol{\theta}, \boldsymbol{\phi}, \mathbf{X}) = \log p_{\boldsymbol{\theta}}(\mathbf{X}, \mathbf{Z}^{(\mathbb{T})}) + \mathcal{H}[q_\phi^{\mathbb{T}}(\mathbf{Z}|\mathbf{X})],
\tag{44}
$$

where $\mathbf{Z}^{(\mathbb{T})}$ denotes the tree-structured reparameterization for latent variables. The first term can be directly computed by

$$
\log p_{\boldsymbol{\theta}}(\mathbf{X}, \mathbf{Z}^{(\mathbb{T})}) = \sum_{i=1}^{N} \log p_{\boldsymbol{\theta}}(\mathbf{x}_i|\mathbf{z}_i^{(\mathbb{T})}) + \log p(\mathbf{z}_i^{(\mathbb{T})}),
\tag{45}
$$

where $\mathbf{z}_i^{(\mathbb{T})}$ is the reparameterization for latent variable $\mathbf{z}_i$, $i = 1, \cdots, N$. And the entropy of the tree-structured posterior $q_\phi^{\mathbb{T}}(\mathbf{Z}|\mathbf{X})$ can be factorized as entropy terms with respect to singleton posterior $q_\phi(\mathbf{z}_i|\mathbf{x}_i)$ and pairwise posterior $q_\phi(\mathbf{z}_i, \mathbf{z}_j|\mathbf{x}_i, \mathbf{x}_j)$

$$
\mathcal{H}[q_\phi^{\mathbb{T}}(\mathbf{Z}|\mathbf{X})] = \sum_{i \in \mathcal{V}} \mathcal{H}[q_\phi(\mathbf{z}_i|\mathbf{x}_i)] + \sum_{(i,j) \in \mathcal{E}} \mathcal{H}[q_\phi(\mathbf{z}_i, \mathbf{z}_j|\mathbf{x}_i, \mathbf{x}_j)] - \mathcal{H}[q_\phi(\mathbf{z}_i|\mathbf{x}_i)] - \mathcal{H}[q_\phi(\mathbf{z}_i|\mathbf{x}_i)].
\tag{46}
$$

where the entropy of singleton posterior $q_\phi(\mathbf{z}_i|\mathbf{x}_i) = \mathcal{N}(\mathbf{z}_i; \boldsymbol{\mu}_i, \mathrm{diag}(\boldsymbol{\sigma}_i^2))$ is given by

$$\mathcal{H}[q_\phi(\mathbf{z}_i|\mathbf{x}_i)] = -\mathbb{E}_{\mathbf{z}_i \sim q_\phi(\mathbf{z}_i|\mathbf{x}_i)}[\log q_\phi(\mathbf{z}_i|\mathbf{x}_i)] \tag{47}$$

$$= -\mathbb{E}_{\mathbf{z}_i \sim q_\phi(\mathbf{z}_i|\mathbf{x}_i)} \left[ \log \left[ (2\pi)^{-\frac{D}{2}} |\boldsymbol{\Sigma}_i|^{-\frac{1}{2}} \exp \left( -\frac{1}{2}(\mathbf{z}_i - \boldsymbol{\mu}_i)^\top \boldsymbol{\Sigma}_i^{-1}(\mathbf{z}_i - \boldsymbol{\mu}_i) \right) \right] \right] \tag{48}$$

$$\overset{\star}{=} \frac{D}{2}[1 + \log(2\pi)] + \frac{1}{2} \log |\boldsymbol{\Sigma}_i| \tag{49}$$

$$= \frac{D}{2}[1 + \log(2\pi)] + \frac{1}{2} \sum_{d=1}^{D} \log \sigma_{id}^2, \tag{50}$$

where $\boldsymbol{\Sigma}_i = \mathrm{diag}(\boldsymbol{\sigma}_i^2)$, and the step $\star$ relies on several properties of the trace operator:

$$\mathbb{E}_{\mathbf{z}_i} \left[ (\mathbf{z}_i - \boldsymbol{\mu}_i)^\top \boldsymbol{\Sigma}_i^{-1}(\mathbf{z}_i - \boldsymbol{\mu}_i) \right] = \mathbb{E}_{\mathbf{z}_i} \left[ \mathrm{tr} \left( (\mathbf{z}_i - \boldsymbol{\mu}_i)^\top \boldsymbol{\Sigma}_i^{-1}(\mathbf{z}_i - \boldsymbol{\mu}_i) \right) \right] \tag{51}$$

$$= \mathbb{E}_{\mathbf{z}_i} \left[ \mathrm{tr} \left( \boldsymbol{\Sigma}_i^{-1}(\mathbf{z}_i - \boldsymbol{\mu}_i)^\top(\mathbf{z}_i - \boldsymbol{\mu}_i) \right) \right] \tag{52}$$

$$= \mathrm{tr} \left( \boldsymbol{\Sigma}_i^{-1} \mathbb{E}_{\mathbf{z}_i} \left[ (\mathbf{z}_i - \boldsymbol{\mu}_i)^\top(\mathbf{z}_i - \boldsymbol{\mu}_i) \right] \right) \tag{53}$$

$$= \mathrm{tr}(\boldsymbol{\Sigma}_i^{-1}\boldsymbol{\Sigma}) = \mathrm{tr}(\mathbf{I}_D) = D. \tag{54}$$

And the entropy of pairwise posterior $q_\phi(\mathbf{z}_i, \mathbf{z}_j|\mathbf{x}_i, \mathbf{x}_j) = \mathcal{N}(\mathbf{z}_i, \mathbf{z}_j; \boldsymbol{\mu}_{ij}, \boldsymbol{\Sigma}_{ij})$ with mean $\boldsymbol{\mu}_{ij} = [\boldsymbol{\mu}_i, \boldsymbol{\mu}_j]^\top$ and covariance matrix

$$\boldsymbol{\Sigma}_{ij} = \begin{bmatrix} \boldsymbol{\sigma}_i \odot \boldsymbol{\sigma}_i & \boldsymbol{\gamma}_{ij} \odot \boldsymbol{\sigma}_i \odot \boldsymbol{\sigma}_j \\ \boldsymbol{\gamma}_{ij} \odot \boldsymbol{\sigma}_i \odot \boldsymbol{\sigma}_j & \boldsymbol{\sigma}_j \odot \boldsymbol{\sigma}_j \end{bmatrix} \tag{55}$$

is similarly given by

$$\mathcal{H}[q_\phi(\mathbf{z}_i, \mathbf{z}_j|\mathbf{x}_i, \mathbf{x}_j)] = -\mathbb{E}_{\mathbf{z}_i, \mathbf{z}_j}[\log q_\phi(\mathbf{z}_i, \mathbf{z}_j|\mathbf{x}_i, \mathbf{x}_j)] \tag{56}$$

$$= -\mathbb{E}_{\mathbf{z}_{ij}} \left[ \log \left[ (2\pi)^{-D} |\boldsymbol{\Sigma}_{ij}|^{-\frac{1}{2}} \exp \left( -\frac{1}{2}(\mathbf{z}_{ij} - \boldsymbol{\mu}_{ij})^\top \boldsymbol{\Sigma}_{ij}^{-1}(\mathbf{z}_{ij} - \boldsymbol{\mu}_{ij}) \right) \right] \right] \tag{57}$$

$$= D + D\log(2\pi) + \frac{1}{2} \log |\boldsymbol{\Sigma}_{ij}| \tag{58}$$

$$= D + D\log(2\pi) + \frac{1}{2} \sum_{d=1}^{D} \left[ \log(1 - \gamma_{ijd}^2) + \log \sigma_{id}^2 + \log \sigma_{jd}^2 \right] \tag{59}$$

where we denote $\mathbf{z}_{ij} = [\mathbf{z}_i, \mathbf{z}_j]^\top$.

### C.2  Evidence Lower Bound for MTreeVI

For a mixture-of-trees structured posterior approximation with respect to the mixture of tree components $\mathbb{MT} = \{\mathbb{T}_1, \cdots, \mathbb{T}_M\}$, the evidence lower bound of our proposed MTreeVI is given by

$$\mathcal{L}^{\mathrm{MT}}(\boldsymbol{\theta}, \phi, \mathbf{X}) = \sum_{m=1}^{M} \pi_m \left[ \log p_{\boldsymbol{\theta}}(\mathbf{X}, \mathbf{Z}^{(\mathbb{T}_m)}) + \mathcal{H}[q_\phi^{\mathbb{T}_m}(\mathbf{Z}|\mathbf{X})] \right], \tag{60}$$

where $\mathbf{Z}^{(\mathbb{T}_m)}$ denotes the tree-structured reparameterization for latent variables with respect to the $m$-th tree component $\mathbb{T}_m, m = 1, \cdots, M$. The first weighted summations can be directly computed by

$$\sum_{m=1}^{M} \pi_m \log p_{\boldsymbol{\theta}}(\mathbf{X}, \mathbf{Z}^{(\mathbb{T}_m)}) = \sum_{m=1}^{M} \pi_m \sum_{i=1}^{N} \log p_{\boldsymbol{\theta}}(\mathbf{x}_i|\mathbf{z}_i^{(\mathbb{T}_m)}) + \log p(\mathbf{z}_i^{(\mathbb{T}_m)}), \tag{61}$$

where $\mathbf{z}_i^{(\mathbb{T}_m)}$ is the reparameterization for latent variable $\mathbf{z}_i, i = 1, \cdots, N$ with respect to the $m$-th tree component, $m = 1, \cdots, M$. And the weighted summations of the entropy terms can be similarly

factorized and calculated as Appendix C.1 by entropy terms with respect to singleton posteriors $q_\phi(\mathbf{z}_i|\mathbf{x}_i)$ and pairwise posteriors $q_\phi(\mathbf{z}_i, \mathbf{z}_j|\mathbf{x}_i, \mathbf{x}_j)$.

## D  Constrained Optimization Details

### D.1  Proof of Proposition 1

*Recall that the spectral radius $r(\cdot)$ is the largest absolute eigenvalue of a matrix. The following show characterizations for acyclicity of the undirected structure induced by the symmetric matrix $\mathbf{A}$:*

1. *Suppose that $\mathbf{A} \in \{0,1\}^{N \times N}$ and $r(\mathbf{A}) < 1$, then $\mathbf{A}$ is acyclic if and only if*

$$\operatorname{tr} \mathbf{A}(\mathbf{I}_N - \mathbf{A}^2)^{-1} = 0. \tag{62}$$

2. *Suppose that $\mathbf{A} \in \{0,1\}^{N \times N}$, then $\mathbf{A}$ is acyclic if and only if*

$$\operatorname{tr} \mathbf{A} \exp(\mathbf{A}^2) = 0. \tag{63}$$

3. *Suppose that $\mathbf{A} \in \mathbb{R}^{N \times N}$, then $\mathbf{A}$ is acyclic if and only if*

$$\operatorname{tr}(\mathbf{A} \odot \mathbf{A}) \exp[(\mathbf{A} \odot \mathbf{A})^2] = 0, \tag{64}$$

*where $\operatorname{tr}(\cdot)$ and $\exp(\cdot)$ represent the trace and exponential of a matrix respectively, and $\odot$ is the Hadamard product.*

*Proof.* The proof relies on the fact that $\operatorname{tr} \mathbf{A}^k$ counts the number of length-$k$ closed walks in a directed graph.

1. Clearly the directed graph induced by the symmetric matrix $\mathbf{A}$ will only have self-loops, and hence $\operatorname{tr} \mathbf{A}^{2k+1} = 0$ for all $k = 1, \cdots, \infty$. In other words, $\mathbf{A}$ has no cycles if and only if $f(\mathbf{A}) = \sum_{k=1}^\infty \sum_{i=1}^N (\mathbf{A}^{2k+1})_{ii} = 0$, then

$$\operatorname{tr} \mathbf{A}(\mathbf{I}_N - \mathbf{A}^2)^{-1} = \operatorname{tr} \sum_{k=1}^\infty \mathbf{A}^{2k+1} = \sum_{k=1}^\infty \sum_{i=1}^N (\mathbf{A}^{2k+1})_{ii} = 0. \tag{65}$$

   The desired result follows.

2. Similar to the proof of Proposition 1.1 by noting that $\mathbf{A}$ has no cycles if and only if $(\mathbf{A}^{2k+1})_{ii} = 0$ for all $k \geq 1$ and all $i \in \{1, \cdots, N\}$, which is true if and only if

$$\operatorname{tr} \mathbf{A} \exp(\mathbf{A}^2) = \operatorname{tr} \sum_{k=1}^\infty \frac{1}{k!} \mathbf{A}^{2k+1} = \sum_{k=1}^\infty \sum_{i=1}^N \frac{1}{k!} (\mathbf{A}^{2k+1})_{ii} = 0. \tag{66}$$

3. The proof is similar to Proposition 1.2 by replacing $\mathbf{A}$ with $\mathbf{A} \odot \mathbf{A}$, which counts weighted closed walks.

$\square$

### D.2  Greedy Searched Initialization

The constrained optimization of tree structure requires an initialization by construction from data, and the easiest way of tree construction is to randomly build from the fully-connected graph by using depth-first-search (DFS) algorithm. Algorithm 1 shows the DFS algorithm for our tree initialization. In the algorithm, $RC_{[\cdot]}$ means randomly choosing one index according to the indicator function; $ID_{[\cdot]}$ represents the set of node indexes satisfying the indicator condition and $\mathcal{N}(i)$ denotes the neighbors of node $i$.

---
**Algorithm 1** DFS Algorithm for Tree Generation
---
**Input:** Fully-connected graph $\mathbb{G}$; number of trees $M$
**Output:** Edges list of generated trees $\boldsymbol{E}$
 1: **procedure** TreeGeneration($M$)                    ▷ Input: #tree $M$
 2:     $\boldsymbol{E} = [\,]$                           ▷ Initial edges list
 3:     **for** $k = 0, \cdots, M-1$ **do**
 4:         $\boldsymbol{V} = [False]^{|\mathcal{V}|}$    ▷ Visited node list
 5:         **while** $False$ in $\boldsymbol{V}$ **do**
 6:             $i \leftarrow RC_{[V==False]}$            ▷ Choose node
 7:             $\boldsymbol{Q} = [i]$                     ▷ Initial queue
 8:             **while** $len(\boldsymbol{Q}) > 0$ **do**
 9:                 $i \leftarrow \boldsymbol{Q}[0]$
10:                 $\boldsymbol{V}[i] \leftarrow True$
11:                 $\boldsymbol{N} = ID_{[V[\mathcal{N}(i)==False]]}$
12:                 **if** $len(\boldsymbol{N}) == 0$ **then**
13:                     $POP(\boldsymbol{Q}, -1)$
14:                     break
15:                 **end if**
16:                 $j \leftarrow RC_{[\boldsymbol{N}]}$    ▷ Choose neighbor
17:                 $\boldsymbol{V}[j] \leftarrow True$
18:                 $APPEND(\boldsymbol{Q}, j)$
19:                 $APPEND(\boldsymbol{E}, [i, j])$
20:             **end while**
21:         **end while**
22:     **end for**
23: **end procedure**
---

To enrich the constructed spanning tree with neighboring correlation information, the uniform sampling process (line 16 in Algorithm 1) in the DFS algorithm can be further modified to generate a meaningful neighborhood for each data instance, by assigning the probability of sampling neighbor $j$ of instance $i$ as

$$\frac{\exp(\cos(\mathbf{x}_j^\top \mathbf{x}_i)/\alpha)}{\sum_{k \in \mathcal{N}(i}\exp(\cos(\mathbf{x}_k^\top \mathbf{x}_i)/\alpha)}, \tag{67}$$

where $\alpha$ is the temperature parameter controlling the trade-off between the precision and diversity of edges, and we find the best configuration of $\alpha$ on the validation set with the values in $\{0.1, 0.2, \cdots, 1.0\}$.

In Table 5, we show the constrained clustering accuracies on the MNIST dataset, with different initializations, and with or without our constrained optimization. It can be seen that without constrained optimization, both initializations underperform. And with constrained optimization, both initializations converge to substantially better performances, with the greedy-searched initialization slightly better. Overall, the constrained optimization procedure makes our proposed TreeVI less sensitive to initializations.

Table 5: Constrained clustering accuracies (%) on the MNIST dataset, with random tree or greedy-searched tree initializations, with or without our constrained optimization

|         | Random Tree | Greedy Search |
| ------- | ----------- | ------------- |
| w/o CO  | 96.58       | 96.70         |
| w/ CO   | 97.31       | 97.45         |

### D.3 Solving Constrained Optimization

In section 2.3, we establish a continuous characterization of acyclicity, leading to the following equality-constrained program (ECP)

$$\min_{\mathbf{A}\in\mathbb{R}^{N\times N}} \quad \ell(\mathbf{A})$$
$$\text{subject to} \quad h(\mathbf{A}) = 0, \tag{68}$$

Solving the constrained optimization requires classical techniques from the mathematical optimization literature. However, this is a nonconvex program since $\{\mathbf{A} : h(\mathbf{A}) = 0\}$ is a non-convex constraint, and hence we are aiming to find its stationary points with non-convex optimization. The algorithm for solving Eq. (68) consists of three steps: ($i$) converting the constrained problem into a series of unconstrained subproblems, ($ii$) optimizing the unconstrained subproblems, and ($iii$) thresholding. The full algorithm is outlined in Algorithm 2.

---

**Algorithm 2** Algorithm for Constrained Optimization

---

**Input:** Initial guess $(\mathbf{A}_0, \alpha_0)$; progress rate $c \in (0, 1)$; tolerance $\epsilon > 0$; threshold $\omega > 0$
**Output:** Threshold matrix $\mathbf{A}$
 1: **for** $t = 0, 1, 2, \cdots$ **do**
 2:      Solve $\mathbf{A}_{t+1} \leftarrow \arg\min_{\mathbf{A}} L^\rho(\mathbf{A}, \alpha_t)$ with $\rho$ such that $h(\mathbf{A}_{t+1}) < ch(\mathbf{A}_t)$
 3:      $\alpha_{t+1} \leftarrow \alpha_t + \rho h(\mathbf{A}_{t+1})$                                  ▷ Dual ascent
 4:      **if** $h(\mathbf{A}_{t+1}) < \epsilon$ **then**
 5:          $\widehat{\mathbf{A}} \leftarrow \mathbf{A}_{t+1}$
 6:          break
 7:      **end if**
 8: **end for**
 9: $\mathbf{A} \leftarrow \widehat{\mathbf{A}} \odot \mathbf{1}(|\widehat{\mathbf{A}}| > \omega)$

---

**Converting constrained optimization into unconstrained subproblems.** The augmented Lagrangian method [28] can be leveraged to solve the equality-constrained program Eq. (68) by first augmenting the original problem with a quadratic penalty:

$$\min_{\mathbf{A}\in\mathbb{R}^{N\times N}} \quad \ell(\mathbf{A}) + \frac{\rho}{2}|h(\mathbf{A})|^2$$
$$\text{subject to} \quad h(\mathbf{A}) = 0, \tag{69}$$

with a penalty parameter $\rho > 0$, which approximates well the solution of the original constrained problem by the solution of unconstrained problems without increasing $\rho$ to infinity. Then the algorithm implements dual ascent for Eq. (69) by defining a dual function with Lagrange multiplier $\alpha$

$$D(\alpha) = \min_{\mathbf{A}\in\mathbb{R}^{N\times N}} L^\rho(\mathbf{A}, \alpha), \tag{70}$$

where $L^\rho(\mathbf{A}, \alpha) = \ell(\mathbf{A}) + \frac{\rho}{2}|h(\mathbf{A})|^2 + \alpha h(\mathbf{A}) \tag{71}$

is the augmented Lagrangian. And the goal is to find a local solution to the dual problem

$$\max_{\alpha\in\mathbb{R}} D(\alpha). \tag{72}$$

Let $\mathbf{A}_\alpha^*$ be the local minimizer of the Lagrangian (70) at $\alpha$, i.e., $D(\alpha) = L^\rho(\mathbf{A}_\alpha^*, \alpha)$. Since the dual objective $D(\alpha)$ is linear in $\alpha$ with the derivative simply given by $\nabla D(\alpha) = h(\mathbf{A}_\alpha^*)$, one can perform dual gradient ascent to optimize the dual problem (72):

$$\alpha \leftarrow \alpha + \rho h(\mathbf{A}_\alpha^*). \tag{73}$$

**Solving the unconstrained subproblem.** The augmented Lagrangian converts the constrained problem (69) into a series of subproblems (70), and our goal is to solve these subproblems efficiently. Let $\boldsymbol{a} = \text{vec}(\mathbf{A}) \in \mathbf{R}^p$, with $p = N^2$. The unconstrained subproblem (70) can be considered as a

typical minimization problem over real vectors:

$$\min_{\boldsymbol{a} \in \mathbb{R}^p} f(\boldsymbol{a}), \tag{74}$$

$$\text{where } f(\boldsymbol{a}) = \ell(\mathbf{A}) + \frac{\rho}{2} |h(\mathbf{A})|^2 + \alpha h(\mathbf{A}) \tag{75}$$

is a smooth objective, for which a number of efficient numerical algorithms are available, such as L-BFGS [5]. In our experiments, the values of $\boldsymbol{a}$ can be learned by a neural network that encodes pairwise correlation.

**Thresholding.** Motivated by post-processing estimates of coefficients via hard thresholding, we threshold the edge weights as follows: after obtaining a stationary point $\widehat{\mathbf{A}}$ of (69) given a fixed threshold $\omega > 0$, set any weights smaller than $\omega$ in absolute value to zero. This strategy also has the effect of "rounding" the numerical solution of the augmented Lagrangian (69), since the solution satisfies $h(\widehat{\mathbf{A}}) \leq \epsilon$ for some all tolerance $\epsilon$ (set to $\epsilon = 10^{-8}$ in our experiments) instead of $h(\widehat{\mathbf{A}}) = 0$ strictly due to numerical precisions. However, a small threshold $\omega$ suffices to rule out cycle-inducing edges since $h(\widehat{\mathbf{A}})$ explicity quantifies the acyclicity of $\widehat{\mathbf{A}}$. Following [45], a fixed value of threshold $\omega = 0.3$ is set in all our experiments.

# E Experimental Details

## E.1 Datasets

The datasets used in the experiments are the followings:

- **MovieLens 20M:** A dataset describing ratings and free-text tagging activities from Movie-Lens, a movie recommendation service. It contains 20,000,263 ratings and 465,564 tag applications across 27,278 movies created by 138,493 users [9].

- **Epinions:** A dataset that records ratings and trust statements issued by users from Epinions, a consumers opinion site where users can review items and assign them numeric ratings in the range 1 to 5, and also express their Web of Trust by issuing trust statements. It consists of 49,290 users who rated a total of 139,738 different items at least once. The total number of review is 664,824. The total number of issued trust statements is 487,181.

- **MNIST:** It consists of 70,000 handwritten digits. The images are centered and of size 28 by 28 pixels, each reshaped to a 784-dimensional vector [21].

- **Fashion MNIST:** A dataset of Zalando's article images consisting of a training set of 60,000 examples and a test set of 10,000 examples [40].

- **Reuters:** It contains 810,000 English news stories [22]. Following the work of [41], we used 4 root categories: corporate/industrial, government/social, markets, and economics as labels and discarded all documents with multiple labels, which results in a 685,071-article dataset. We computed tf-idf features on the 2000 most frequent words to represent all articles. A random subset of 10,000 documents is then sampled.

- **STL-10:** It contains color images of 96-by-96 pixel size. There are 10 classes with 13,000 examples each [7]. As pre-processing, we extracted features from the STL-10 image dataset using a ResNet-50 [10], as in previous works [15].

## E.2 Implementation Details

**Synthetic Data.** For synthetic VAEs, we apply a two-layer feed-forward neural network for the generative model $p_{\boldsymbol{\theta}}(\mathbf{x}_i|\mathbf{z}_i)$ and a two-layer feed-forward neural network for the variational posterior approximation $q_{\boldsymbol{\phi}}(\mathbf{z}_i|\mathbf{x}_i)$, with each $\mathbf{z}_i \in \mathbb{R}^D$ where $D = 4$. The traditional mean-field approximation assumes the posterior distribution for each latent to be fully factorized: $q_{\boldsymbol{\phi}}(\mathbf{z}_i|\mathbf{x}_i) = \prod_{d=1}^{D} q_{\boldsymbol{\phi}}(z_{id}|x_{id})$, while in our methods the posterior is assumed to be tree-structured and mixture-of-trees structured, which can be factorized into singleton posterior $q_{\boldsymbol{\phi}}(z_{id}|x_{id})$ and pairwise posterior $q_{\boldsymbol{\phi}}(z_{id}, z_{ie}|x_{id}, x_{ie})$ both assumed to be Gaussian. Each edge in the tree structure corresponds to a correlation parameter $\gamma_{de}$ to be learned in the bi-variate normal distribution, which

Table 6: Hyperparameters setting of constrained clustering task.

|  | MNIST | fMNIST | Reuters | STL-10 |
|---|---|---|---|---|
| Batch size | 256 | 256 | 256 | 256 |
| Epchs | 1000 | 500 | 500 | 500 |
| Learning rate | 0.001 | 0.001 | 0.001 | 0.001 |
| Decay | 0.9 | 0.9 | 0.9 | 0.9 |
| Epochs decay | 20 | 20 | 20 | 20 |

is shared for all data points to capture dimension-level correlation. Our synthetic VAE is trained for 200 epochs for all methods. We apply stochastic gradient optimizations with a step size of 0.005, and use the Adam algorithm [17] to adjust the learning rates.

**Constrained Clustering.** To implement our methods, we are careful in maintaining a fair comparison with the baseline methods. In particular, we adopt the same encoder and decoder feed-forward architecture used by the baseline method: four layers of 500, 500, 2000, $D$ units respectively, where $D = 10$ unless stated otherwise. VAE-based baselines and the VAEs equipped with our posterior approximation methods are pretrained for 10 epochs while the DEC-based baselines involve 50 epochs of pretraining for each layer and 100 epochs of pretraining as finetuning. Each dataset is divided into training and test sets, where the former one is used for training and our reported results are operated on the latter one. The pairwise constraints used in our experiments are randomly chosen within the training set, by randomly sampling any two data points and assigning a must-link if they have the same label and a cannot-link otherwise. The absolute values of the elements $|W_{ij}|$ in the pairwise prior information matrix are set to $10^4$ for all datasets for convenience, and 6000 pairwise constraints are sampled for training both our methods and other baselines. Following DC-GMM [24], the hyper-parameters are universally set for four different datasets, as shown in Table 6. The learning rate is set to 0.001 and it decreases every 20 epochs with a decay rate of 0.9. The number of tree components adopted in our MTreeVI is set to $M = 3$.

**User Matching & Link Prediction.** For both tasks of user matching and link prediction, we set the dimensionality of latent embeddings as $D = 100$ for all methods. For VAE-based methods (including TreeVI and MTreeVI), we apply a two-layer feed-forward neural network for the generative model $p_\theta(\mathbf{x}_i|\mathbf{z}_i)$ and a two-layer feed-forward neural network for the posterior approximation $q_\phi(\mathbf{z}_i|\mathbf{x}_i)$. The model likelihood function $p_\theta(\mathbf{X}|\mathbf{Z})$ is a multinomial distribution, and the singleton posterior distributions $q_\phi(\mathbf{z}_i|\mathbf{x}_i)$ are all diagonal normal distributions. For CVAE$_{\text{corr}}$ and our methods, we also learn a two-layer feed-forward neural network that takes the concatenation $[\mathbf{x}_i; \mathbf{x}_j]$ as input and output the correlation parameter between $\mathbf{z}_i$ and $\mathbf{z}_j$ on each of the $D$ dimensions. To determine the tree structure for TreeVI and MTreeVI, we take average of these correlation parameters across all $D$ dimensions and obtain a correlation matrix $\mathbf{A} \in \mathbb{R}^{N \times N}$ or a set of correlation matrices that is used for constrained optimization. For GraphSAGE, we choose to use $K = 2$ aggregation steps and use the mean aggregator function. We use $Q = 20$ negative samples to optimize the loss function. For all methods, we apply stochastic gradient optimizations with a step size of 0.001, and use the Adam optimizer to adjust the learning rates. The number of tree components adopted in our MTreeVI is set to $M = 3$ for both tasks.

### E.3 Further Experiments

**Synthetic Data.** We design a synthetic dataset with a graph-structured latent variable model. The dataset contains $N = 6000$ data points $\mathbf{x}_1, \cdots, \mathbf{x}_N \in \mathbb{R}^D$ with $D = 4$, each independently generated from the conditional distribution $p(\mathbf{x}|\mathbf{z}) = \mathcal{N}(\mathbf{x}; \boldsymbol{\theta}\mathbf{z}, \sigma^2 \mathbf{I}_4)$ given latent embeddings $\mathbf{z}_1, \cdots, \mathbf{z}_N \in \mathbb{R}^D$ where $\sigma^2$ is a fixed value and set to 0.5. The latent embeddings $\mathbf{z}_1, \cdots, \mathbf{z}_N$ are drawn from a standard normal distribution $p(\mathbf{z}) = \mathcal{N}(\mathbf{z}; \mathbf{0}_4, \mathbf{I}_4)$, and the graph-structured correlation

Table 7: Estimated lower bounds (ELBO) of VAE with posterior distributions approximation by mean-field, tree-structured, and mixture-of-trees distributions, compared to ground truth log-likelihood $\log p(\mathbf{X})$.

| Methods | Lower Bound |
|---------|-------------|
| Mean-field | -9.022 |
| TreeVI (1 Tree) | -8.8838 |
| MTreeVI (2 Trees) | -8.8729 |
| MTreeVI (3 Trees) | -8.8690 |
| $\log p(\mathbf{X})$ | -8.8142 |

Table 8: Clustering with constructed affinity matrices performances (%) of our proposed methods TreeVI and MTreeVI compared with baselines

| | MNIST | | | fMNIST | | | Reuters | | | STL-10 | | |
|---|---|---|---|---|---|---|---|---|---|---|---|---|
| | ACC | NMI | ARI | ACC | NMI | ARI | ACC | NMI | ARI | ACC | NMI | ARI |
| VaDE | 92.00 | 90.80 | 82.90 | 63.20 | 62.90 | 48.30 | 83.35 | 63.10 | 67.09 | 84.45 | 79.10 | 73.98 |
| DEC | 87.93 | 80.71 | 78.35 | 60.61 | 63.70 | 47.41 | 76.33 | 51.91 | 57.31 | 78.56 | 72.47 | 63.91 |
| DCN | 83.10 | 82.50 | 75.60 | 53.10 | 57.10 | 38.60 | 76.60 | 61.10 | 65.40 | 78.40 | 65.50 | 60.70 |
| DGG | 97.58 | 92.50 | 93.40 | 81.33 | 73.20 | 67.90 | 93.50 | 81.23 | 87.85 | **90.59** | **81.58** | **79.80** |
| DC-GMM | 95.28 | 90.01 | 90.55 | 77.24 | 70.05 | 65.73 | 92.38 | 80.68 | 88.05 | 88.70 | 78.90 | 77.50 |
| TreeVI | 96.30 | 92.21 | 92.82 | 78.90 | 72.08 | 66.39 | 95.12 | 82.40 | 89.02 | 90.22 | 80.97 | 79.30 |
| MTreeVI | **97.66** | **92.96** | **93.52** | **81.57** | **73.61** | **68.20** | **95.31** | **82.64** | **89.20** | 90.50 | 81.22 | 79.62 |

is incorporated into the sample generation via a coefficient matrix $\boldsymbol{\theta}$

$$\boldsymbol{\theta} = \begin{bmatrix} 2 & 0 & 1 & 0 \\ 1 & 2 & 0 & 0 \\ 0 & 1 & 2 & 0 \\ 0.3 & 0.3 & 0.4 & 2 \end{bmatrix} \tag{76}$$

And we adopt different number of tree components to model the posterior correlation, the comparison of their estimated evidence lower bounds to the ground truth log-likelihood is presented in Table 7. It can be observed that tree-structured poterior benefits from capturing more pairwise correlations, and the mixture of more tree components better approximates the ground truth log-likelihood.

**Clustering with Constructed Affinity Matrix.** Besides the weak supervision of pairwise constraints, we also analyze the unsupervised scenario of constructed affinity matrix in this work, where we construct an affinity matrix under a specific similarity measure and each entry is a value between $[0, 1]$ reflecting the similarity between a sample pair. To construct the affinity matrix $\mathbf{W}$, we find a set of nearest neighbors for a given data point and compute their similarity using a predefined kernel function, such as Gaussian kernel

$$\mathbf{W}_{ij} = \begin{cases} \exp\left(-\frac{\|\mathbf{x}_i - \mathbf{x}_j\|_2^2}{2s_i^2}\right), & \text{if } \mathbf{x}_j \in \mathcal{N}(\mathbf{x}_i) \\ 0, & \text{otherwise} \end{cases} \tag{77}$$

where $s_i$ is a predefined scalar, $\mathcal{N}(\mathbf{x}_i)$ denotes the set consisting of the nearest $N_s$ neighbors of $\mathbf{x}_i$. To enhance robustness to different datasets, we train a Siamese network to measure the similarity between data points. With the constructed affinity matrix $\mathbf{W}$, the conditional distribution of the cluster assignment can be similarly defined as Eq. (19), with the weighting function modified as

$$h_i(\mathbf{c}, \mathbf{W}) = \prod_{j \neq i} \exp\left[\mathbf{W}_{ij}\delta_{c_i c_j} + (1 - \mathbf{W}_{ij})(1 - \delta_{c_i c_j})\right]. \tag{78}$$

The clustering performances of our proposed methods against baseline methods are shown in Table 8.

### E.4  Resource Usage

Experiments were conducted on an internal computing cluster. Each experiment configuration used one NVIDIA GPU (either a 2080TI or 3090TI), 16 CPUs and a total of 24GB of memory.

