# OpenReview forum: "TreeVI: Reparameterizable Tree-structured Variational Inference for Instance-level Correlation Capturing"
_NeurIPS.cc/2024/Conference — NeurIPS 2024 poster_

### Official Review · Reviewer_smr5 · 2024-07-11

**Soundness:** 3
**Presentation:** 2
**Contribution:** 2
**Rating:** 6
**Confidence:** 3

**Summary:**

The paper introduces a new type of variational inference that allows correlations across training samples in the form of a tree, making it suitable for applications with graph-structured data or explicit constraints. The method extends to multiple trees for capturing more complex correlations and includes a mechanism for automatically learning the underlying tree structures from data. Experimental results demonstrate that the proposed approach provide advantages in settings such as constrained clustering, user rating and link prediction.

**Strengths:**

While using tree structures in VAEs have been already explored [1], the setting used by this paper deviates from [1] as it considers correlations across samples and not across hierarchical latent variables. I believe that the idea is novel and the method sound. The experiments show improvements over related work and show a broad applicability of the proposed inference.

[1] Manduchi et al. Tree Variational Autoencoders, NeurIPS 2023, https://proceedings.neurips.cc/paper_files/paper/2023/file/ac58b418745b3e5f10c80110c963969f-Paper-Conference.pdf.

**Weaknesses:**

1. I find the setting of the paper very confusing, I worked on hierarchical VAEs and therefore had some difficulty in understanding the notation of the paper. Figure 1 did not help as notations are misleading: what do the numbers in the nodes represent? Why in GraphVAE the nodes are labeled z_i while in the trees they are labeled with just a number? I believe the paper would be stronger if (a) notations could be better explained and (b) if comparison with TreeVAE [1] could be explained to differentiate the two settings.
2. I also believe that the paper could benefit from a more thorough analysis of the computational complexity  of the proposed appraoch(there is only an empirical comparison with DC-GMM).
3. Experiments are limited to simple datasets, standard deviations are not reported in the table, which I believe are of crucial importance especially in clustering (given the high instability of the model with different initialisation), and few experimental settings are explained. E.g. At what value is M set in the experiments?
4. The construction of the trees, while interesting, have not been properly validated. For example, in 4.2 it would be more convincing to provide evidence that the learnt tree are consistent with the instance-level constrains (are similar pair of samples close in the tree?).
5. I think the paper is not always clearly written and it contains several typos (see line 350 or 316, 317 as examples).

**Questions:**

1. Could the authors provide a proof of Equation 4 (in the appendix)?
2. 4.1 Why the tree structures are defined a-priori and not learnt from the data? That would be an interesting experiment to validate the correctness in constructing the tree in the synthetic scenario.
3. Line 317: how do you compute the latent embedding distance? Is it the distance in the tree or the e.g. L2 distance in the embedding space?

**Limitations:**

I believe the authors have addressed the limitations of the proposed approach.

---

> ### Author Rebuttal · Authors · 2024-08-07
>
> We thank the reviewer for the comments and suggestions. Below we respond to each of the raised questions.
>
> **Q1: What do the numbers in the nodes represent in Figure 1?**
>
> A1: Sorry for the confusing notations used in the Figure 1. In Figure 1 (b) and \(c), the numbers 1, 2, ... 5 in the trees should also be written as $\mathbf{z}_1, \mathbf{z}_2, \cdots, \mathbf{z}_5$. In our model, we assume for each data instance $\mathbf{x}_i$, it corresponds to a latent variable $\mathbf{z}_i$, and Fig. 1 (a), (b), \(c) are used to discribe the correlation structures among differnt latent variables $\mathbf{z}_i$. The notations in this paper are different from those in TreeVAE because TreeVAE focuses on how to model the hierarchical structure of different dimensions within one instance, while ours devotes to modeling the correlation among different instances.
>
> **Q2: I also believe that the paper could benefit from a more thorough analysis of the computational complexity of the proposed appraoch.**
>
> A2: Thanks for the suggestions. For amortized variational inference, the computational complexity mainly comes from running the re-parameterization neural network, thus the complexity can be measured by how many runs of the re-parameterization neural network are required. For mean-field VI, each instance only need to pass the neural network once, thus for a dataset with $N$ instances, the number runs of the neural network is $O(N)$.
>
> In our method, by restricting the correlation matrix $\mathbf{R}$ to a form constructed from a tree, we prove that all of the non-zero elements in $\mathbf{L}$ can be explicitly computed from the $|\mathcal{E}|$ parameters $\gamma_{ij}$ for $(i,j)\in \mathcal{E}$. Thus, we only need to re-parameterize the $|\mathcal{E}|$ parameters $\gamma_{ij}$, and run re-parameterization neural network $|\mathcal{E}|$ times, in addition to re-parameterizing the mean and variance, which requires a complexity of $O(N)$. For a tree, it is known that $|\mathcal{E}|\le N$, Thus, the total number of runs of the neural network is approximately $O(2N)$. This is also consistent with our observation in the experiments that the training time of our method is generally 2~3 times of mean-field methods.
>
> **Q3: Standard deviations are not reported in the table, and few experimental settings are explained. E.g. At what value is M set in the experiments?**
>
> A3: We thank the reviewer for pointing out the missing details. The results of constrained clustering with standard deviations are supplemented in our attachment to the global rebuttal, and will be included into our final version.
>
> In our experiments, the value of $M$ is set to 3, and more detailed experimental settings are elaboratred in Appendix E.2. We will relocate some of the settings into the main context of the paper in our final version.
>
> **Q4: In 4.2 it would be more convincing to provide evidence that the learnt tree are consistent with the instance-level constraints.**
>
> A4: Thanks for the suggestion. Our method is mainly used to capture the inherent correlations among different instances. Thus, the correlations captured by the trees are not necessary to be consistent with the constrains, but instead are more inclined to align with the similarities among instances. To validate this, we plot the tree learned over the constrained clustering experiment on MNIST dataset, as shown in the Figure 1 in the attachment of global rebuttal. From the figure, we can see that instances from the same category are connected more tightly than those from different categories in the learned tree. This demonstrates that our method has the abilities to capture the underlying inherent correlations among different instances.
>
> **Q5: Could the authors provide a proof of Equation 4?**
>
> A5: Equation 4 is a result of the Markov tree dependent distributions, for a complete proof we can refer to [1] which is mainly done by induction. By the way, this equation can also be regarded as a special case of the Hammersley-Clifford theorem. We will add these citations in our final version.
>
> Reference
>
> 1. Meeuwissen A.M.H. and Cooke R.M., Tree dependent random variables Report 94-28, Delft University of Technology, 1994.
>
> **Q6: 4.1 Why the tree structures are defined a-priori and not learnt from the data? Validate the correctness of the the trees in the synthetic scenario.**
>
> A6: The experiment in Sec. 4.1 is a toy experiment to illustrate the contribution of each component of our method in capturing the accurate correlation. For better controlability, we choose to use pre-defined trees, instead of trees learned from the data. That is because if we learn the trees from the data, the tree $\mathbb{T}_1$ in MTreeVI($\mathbb{T}_1$, $\mathbb{T}_2$) and the tree $\mathbb{T}_1$ in TreeVI($\mathbb{T}_1$) could be different. Thus, we choose to fix the two trees $\mathbb{T}_1$ and $\mathbb{T}_2$. As seen from the experiment, when using a single tree, either $\mathbb{T}_1$ or $\mathbb{T}_2$, better performance is obained comparing with the mean-field method. When the two trees are used simultaneously, we can see that the performance is further boosted.
>
> Since the 4 dimensions in the toy experiment are all closely correlated, any tree would be good for capturing the correlation among  the 4 dimensions. Thus, it is hard to clearly show which tree is better than the other. But we have demonstrated the consistency between the learned tree and the underlying correlation in the constrained clustering experiments, which is stated in the answer to Q4.
>
> **Q7: How to compute the latent embedding distance**
>
> A7: We utilize the same measurement of latent embedding distance as CVAE for fair comparison, which is computed based on the expected quadratic L2 distance $\mathbb{E}_{q(\mathbf{z}_i,\mathbf{z}_j)}[\| \mathbf{z}_i-\mathbf{z}_j \|^2]$ of the latent distribution for VAE-based methods. Additionally for GraphSAGE, we simply use quadratic L2 distance $\| \mathbf{z}_i-\mathbf{z}_j \|^2$.

---

> > ### Comment · Reviewer_smr5 · 2024-08-09
> >
> > A4: I understand that the learnt correlation should capture the similarity of instances but I would assume it should also reflect the given constraints. Could the authors provide further explanation of why the correlations captured by the tree should not be guided by the pairwise constraints (and as a consequence why the tree should not reflect the given constraints)?

---

> > > ### Author Response · Authors · 2024-08-11
> > >
> > > Thanks for your reply to our rebuttal. The reason why we cannot expect that the links imposed by constraints are largely aligned with the edges in the tree is that for two non-adjacent nodes in a tree, if they are close, they still could be strongly correlated. For example, for a tree with 3 nodes (node 1 connected to node 2, and node 2 connected to node 3), although node 1 is not directly connected to node 3, node 1 and node 3 still could be strongly correlated.
> > >
> > > Thus, for two nodes that are constrained to be linked, it is fine to see they are not directly connected in the tree, as long as they are close to each other in the tree.
> > >
> > > To investigate whether the learned tree has reflected the constraints correctly, we calculate the averaged correlation coefficient between nodes with must-link constraints as well as the averaged coefficient between nodes with cannot-link constraints (still conducted on the MNIST dataset). We observe that the averaged correlation coefficient of nodes with must-link constraints is 0.6481, showing strong positive correlation. On the other hand, the averaged correlation coefficient of nodes with cannot-link constraints is -0.5757, showing strong negative correlation. The results clearly demonstrate that the learned tree reflects the information imposed by the must-link and cannot-link constraints correctly.

---

> > > > ### Comment · Reviewer_smr5 · 2024-08-13
> > > >
> > > > Thank you for the clarification. I raised my score to weak acceptance.

---

### Official Review · Reviewer_96LZ · 2024-07-12

**Soundness:** 3
**Presentation:** 3
**Contribution:** 3
**Rating:** 7
**Confidence:** 3

**Summary:**

The authors propose to use a variational approximation that captures tree-structured correlations as a middle ground between scalable fully-factorized approximations and more expressive approximations which aim to capture the full correlation structure. They show how these tree structured correlations give rise to variational distributions that factorize into tree-structured Markov random fields, whose ancestral sampling procedure informs a matrix-form reparameterization. The authors further extend their methods to mixtures of tree-structured variational distributions and show how to learn tree-structure by jointly optimizing over possible tree-constraint adjacency matrices.

**Strengths:**

Their methods are methodologically interesting, well described. The experiments compare the proposed method to relevant prior work and demonstrate that the method is performing well empirically.

**Weaknesses:**

It is unclear how important the initialization is and if there always exists a straightforward initialization procedure (see related questions). It would be nice to see some result on how the methods perform without initialization.

**Questions:**

The authors report that their method is only about 2-3 times more costly than mean-field VI, however, the maximum number of edges (i.e. additional correlations that have to be learned) in a DAG with $n$ nodes is $\mathcal{O}(n^2)$. So in the worst-case instances the computational cost could still be quite high. Are these instances just not learned in practice or do the authors explicitly regularize to encourage a degree of sparsity in the learned trees? Or is there another reason why this cost is not seen in practice?

- It seems that in order to learn a rich correlation structure initialization is quite important. How well does the proposed method perform without instance specific initialization?

- The initialization itself requires to compute pairwise *neighborhood probabilities* on the data, however, depending on the task it might be more or less clear how to compute meaningful neighborhood probabilities. Is the assumption that we know how to compute distances (or some form of meaningful similarity) between data points explicitly?

- Something small that I noticed. The acronyms ARI (adjusted rand index) and NMI (normalized mutual information) have not been introduced and might not be familiar to readers not familiar with clustering.

--- Edit ---

The authors addressed my concerns regarding the computational complexity (the graph structure is a tree not a DAG) and the importance of the initialization. Consequently I am raising my score to an Accept (7).

**Limitations:**

-

---

> ### Author Rebuttal · Authors · 2024-08-07
>
> We thank the reviewer for the comments and suggestions. Below we respond to each point raised by the reviewer:
>
> **Q1: It is unclear how important the initialization is and if there always exists a straightforward initialization procedure.**
>
> A1: Below we show the constrained clustering accuracies (%) on the MNIST dataset, with different initializations (random tree or greedy-searched tree), with or without our constrained optimization. It can be seen that without constrained optimization, both initializations underperform. And with constrained optimization, both initializations converge to substantially better performances, with the greedy-searched initialization slightly better. Overall, the constrained optimization procedure makes our proposed TreeVI  less sensitive to initializations.
>
> |  | Random Tree | Greedy Search |
> | -------- | -------- | -------- |
> | w/o CO | 96.58 | 96.70 |
> | w/ CO  | 97.31 | 97.45 |
>
> **Q2: The maximum number of edges in a DAG with $n$ nodes is $\mathcal{O}(n^2)$. So in the worst-case instances the computational cost could still be quite high**
>
> A2: In our method, we require to use a tree, instead of a DAG, to capture the correlation among instances. For a tree, the number of edges $|{\mathcal{E}}|$ is at most $N-1$, where $N$ denotes the number of vertices. Thus, the maximum number of edges is $O(N)$, instead of $O(N^2)$. As proved in our paper, by using our method, we only need to re-parameterize the edge parameters $\gamma_{ij}$ for $(i,j)\in {\mathcal{E}}$, and thus only need to run the neural network by $|{\mathcal{E}}|$ times for each epoch, in addition to the runs required by a mean-field method, which is $O(N)$. Thus, the total number of runs of the re-parameterization neural networks is approximately $2N$, roughtly two times of that required by mean-field variational inference. That's why we observed that the training time of our method is generally 2~3 times of the mean-field methods in our experiments.
>
> **Q3: Depending on the task it might be more or less clear how to compute meaningful neighborhood probabilities.**
>
> A3: The probability with respect to each pair of data points is calculated according to the cosine similarities between them, which can be leveraged to generate a meaningful neighborhood for each data instance. For more details we refer to Appendix D.2.
>
> **Q4: Something small that I noticed. The acronyms ARI and NMI have not been introduced and might not be familiar to readers not familiar with clustering.**
>
> A4: Sorry for the missing explanations for the acronyms of clustering metrics. These details will be included into our final version.

---

> > ### Comment · Reviewer_96LZ · 2024-08-12
> > **Re: Rebuttal by Authors**
> >
> > Thank you for addressing my questions. After reading all of the reviews and corresponding rebuttals I'm inclined to raise my score to an Accept (7).
> >
> > Below are some comments and follow-up questions regarding the addressed points.
> >
> > **Q2:** This makes perfect sense—my apologies for my blatant confusion of graph structures.
> >
> > **Q3:** I had another look at Appendix D.2. and believe I understand how the procedure assigns the probability of sampling neighbors based on pairwise cosine similarities. However, while I understand that the cosine similarity can be a useful measure of similarity in certain settings, it is a priori unclear to me why we would expect it to be a good measure of similarity between arbitrary instance embeddings in Euclidean space. In any case, it seems that cosine-similarities are a reasonable first choice and that the procedure can easily be adapted to incorporate other measures of similarity.
> >
> > One more small thing. There is a small typo in Appendix D.2. The neighborhood operator in Equation 67 is missing a closing parentheses.

---

> > > ### Author Response · Authors · 2024-08-12
> > >
> > > Thanks a lot for saying that you are inclined to raise your score to Accept (7). This really means a lot to us, and we hope our response below can convince you to take the final action of raising the score .
> > >
> > > Yes, we cannot guarantee the cosine similarity will always find similar instances, but as evidenced by its successes in a wide range of scenarios, cosine similarity is often not a bad choice when measuring the similarity of instances.
> > >
> > > Moreover, in addition to the greedy-search method, in our paper, we also proposed an optimization method to learn the tree from data, where the greedy search method is only used to provide an initialization for the optimization method. Our experimental results show that even if a random tree is used as the initialization, we can still get a good performance. Thus, with the proposed optimization-based tree learning method, the initialization does not play a very important role.
> > >
> > > Thanks for spotting the missing parentheses for us in Eq. 67, we will correct it in our final version.

---

### Official Review · Reviewer_BYBL · 2024-07-12

**Soundness:** 3
**Presentation:** 2
**Contribution:** 3
**Rating:** 6
**Confidence:** 3

**Summary:**

This paper proposes Tree-structured Variational Inference (TreeVI), a novel method for capturing instance-level correlations in the posterior distribution of latent variables. TreeVI represents correlations between latent variables using a tree structure. This enables reparameterization of latent variables using ancestral sampling, allowing for efficient training. Furthermore, the authors propose Multiple Tree-structured Variational Inference (MTreeVI) to represent more complex correlation structures using multiple tree structures. They also propose a method to learn the tree structure from data. The proposed methods demonstrate superior performance compared to existing approaches on various tasks using both synthetic and real-world datasets.

**Strengths:**

* The approach successfully incorporates inter-instance correlations into the approximate posterior distribution while maintaining efficient training by utilizing a tree structure.
* The proposed methods show improved performance over existing techniques across various tasks.

**Weaknesses:**

In the experiments on real-world data, there is insufficient explanation of what the tree structure represents. The tree structure is used merely as a means to relax the independence assumption of the approximate posterior distribution while maintaining training efficiency. The study could have been enhanced by providing some insights derived from the tree structures obtained through training.

**Questions:**

On line 78, $\mathcal{V} = \{ 1, 2, \cdots , N \}$ is defined. However, in the Synthetic VAE experiment in Section 4.1, despite $N=6000$, $\mathcal{V}= \{ 1,2,3,4 \}$ is used. Could you explain this discrepancy?

**Limitations:**

The limitations are appropriately addressed within the paper, and I have no additional suggestions in this regard.

---

> ### Author Rebuttal · Authors · 2024-08-07
>
> We thank the reviewer for the comments and suggestions. Below we respond to each point raised by the reviewer:
>
> **Q1: The study could be enhanced by providing some insights derived from the tree structures obtained through training.**
>
> A1: Thanks for the suggestions. Our method is mainly used to capture the inherent correlations among different instances. Thus, the learned trees are inclined to reflect the correlation among different instances. To see this, we plot the tree learned over the constrained clustering experiment on MNIST dataset, which is shown in the Figure 1 in the attachment of global rebuttal. From the figure, we can see that instances from the same category are connected more tightly than those from different categories in the learned tree. This demonstrates that our method has the abilities to capture the underlying inherent correlations among different instances. The figure and the explanation to it will be included in our final paper.
>
>
>
> **Q2: Discrepancy of notations used in the synthetic VAE experiment**
>
> A2: The discrepancy arises becuase in this toy experiment, to demonstrate our method is able to catpure correlaitons, we use it to capture correlation among different dimensions within one instance, ranther than capturing correlations among different instances. That is why we see the number of instances $N=6000$, but we only set the vertex set as ${\mathcal{V}}=\{1, 2, 3, 4\}$. That is because the number of dimensions of the latent space is 4 in this toy experiment. We will make this point clearer in our final version.

---

> > ### Comment · Reviewer_BYBL · 2024-08-09
> > **Additional Questions**
> >
> > Thank you for your response.
> >
> > I think the figure in the answer to Q1 is interesting. Including it in the final paper would enhance the value of the paper.
> >
> > However, I still have questions regarding the answer to Q2. Since many equations in the paper are based on the assumption that $\mathcal{V}=\\{1, 2, \dots , N\\}$, it is not clear to me whether it can be directly applied to capturing correlations between dimensions within one instance. For example, if $\mathcal{V}=\\{1, 2, \dots , D\\}$, how would equations (4), (6), (7), and (12) be modified?

---

> > > ### Author Response · Authors · 2024-08-11
> > >
> > > We thank the reviewer for the comments and suggestions. Capturing correlations among dimensions within a specific latent embedding $\mathbf{z}$ can be done by simply replacing the latent variables from $\mathbf{z} _ i$ to $z_d$, $d \in \mathcal{V} = \\{1,\cdots,D\\}$, where $\mathbf{z} _ i \in \mathbb{R}^D$ denotes the latent variable corresponding to instance ${\mathbf{x}} _ i$; and $z_d$ just denotes the $d$-th dimension of latent variable ${\mathbf{z}} \in {\mathbb{R}}^D$. So the expressions of Eq. (4),(6) and (7) will be unchanged, only changing the  meaning of the index $i$ from $i$-th instance to $i$-th dimension. Eq. (12) will be slightly modified. When capturing the correlation among dimensions within one instance, we don't need to feed two instances ${\mathbf{x}} _ i$ and ${\mathbf{x}} _ j$ into the re-parameterization network $f_\phi(\cdot, \cdot)$, but instead only need to feed one instance ${\mathbf{x}}$. Thus, Eq. (12) becomes $\\{\gamma _ {ij}\\} _ {(i,j)\in {\mathcal{E}}} = f _ \phi({\mathbf{x}})$, where here $i$ and $j$ mean the $i$-th and $j$-th dimension; and ${\mathcal{E}}$ denotes the set of edges that connect different dimensions.

---

> > > > ### Comment · Reviewer_BYBL · 2024-08-12
> > > >
> > > > Thank you for your explanation.
> > > >
> > > > > Eq. (4),(6) and (7) will be unchanged, only changing the meaning of the index $i$ from $i$-th instance to $i$-th dimension.
> > > >
> > > > If so, the index $i$ of $\boldsymbol{x}_i$ in Eq. (4) moves in $\\{ 1, 2, \dots , D\\}$. Is it correct?
> > > > Could you write down the modified version of Eq. (4) here?

---

> > > > > ### Author Response · Authors · 2024-08-12
> > > > >
> > > > > Thanks for your reply.
> > > > >
> > > > > > If so, the index $i$ of $\boldsymbol{x}_i$ in Eq. (4) moves in $\{ 1, 2, \dots , D\}$. Is it correct? Could you write down the modified version of Eq. (4) here?
> > > > >
> > > > > Since we are capturing correlations among dimensions within a latent embedding $\mathbf{z}$, we don't need to feed two instances $\mathbf{x} _ i$ and $\mathbf{x} _ j$ as input in Eq. (4), but only one instance $\mathbf{x}$ is needed instead. So the modified version of Eq. (4) should be
> > > > > $$
> > > > > q _ \phi^{\mathbb{T}} (\mathbf{z}|\mathbf{x}) = \prod _ {i \in \mathcal{V}} q_\phi (z _ i|\mathbf{x}) \prod _ {(i,j) \in \mathcal{E}} \frac{q _ \phi (z _ i,z _ j|\mathbf{x})}{q _ \phi (z _ i|\mathbf{x}) q _ \phi (z _ j|\mathbf{x})},
> > > > > $$
> > > > > where $\mathcal{V} = \\{1,\cdots,D\\}$ denotes the set of dimension indices and $\mathcal{E}$ denotes the set of edges that connect different dimensions. Here the index $i$ represents different dimensions of the latent embedding. These modified expressions and subtle details regarding to dimension-level correlations will be included in appendix to make this point clearer in our final version.

---

> ### Comment · Reviewer_BYBL · 2024-08-12
>
> I believe the original equation (4) was the approximate joint posterior distribution of $\boldsymbol{z}\_1, \boldsymbol{z}\_2, \dots , \boldsymbol{z}\_N$ given $\boldsymbol{x}\_1, \boldsymbol{x}\_2, \dots , \boldsymbol{x}\_N$.
> The equation above seems to be for a single instance. Is my understanding correct that the original equation (4) can be written out as follows?
>
> $q\_{\boldsymbol{\phi}}^{\mathbb{T}}(\boldsymbol{Z}|\boldsymbol{X}) = \prod\_{n=1}^N \left( \prod_{i \in \\{ 1, \dots , D \\}} q\_{\boldsymbol{\phi}} (z\_{n,i}|\boldsymbol{x}\_n) \prod\_{(i,j) \in \\{ 1, \dots , D \\}^2} \frac{q\_{\boldsymbol{\phi}}(z\_{n,i}, z\_{n,j}|\boldsymbol{x}\_n)}{q\_{\boldsymbol{\phi}}(z\_{n,i}|\boldsymbol{x}\_n)q\_{\boldsymbol{\phi}}(z\_{n,j}|\boldsymbol{x}\_n)} \right)$,
> where $z_{n,i}$ denotes the $i$-th element of the $n$-th latent embedding $\boldsymbol{z}_n$.

---

> ### Author Response · Authors · 2024-08-12
>
> Yes, your understanding is correct. But the subscript $(i, j)\in \\{1, \cdots, D \\}^2$ need to be written as $(i, j)\in {\mathcal{E}}$ since  only a fraction of instances are connected in the tree.

---

> > ### Comment · Reviewer_BYBL · 2024-08-12
> >
> > So, if we explicitly show the instance subscript $n$, can we say that equations (6), (7), and (12) can be written as follows?
> >
> > $q_{\boldsymbol{\phi}}^\mathbb{T}(\boldsymbol{Z}|\boldsymbol{X}) = \prod_{n=1}^N \left( q_{\boldsymbol{\phi}}(z_{n,1}) \prod_{(i,j) \in \mathcal{E}} q_{\boldsymbol{\phi}} (z_{n,j} | z_{n,i}) \right)$
> >
> > $q_{\boldsymbol{\phi}} (z_{n,j} | z_{n,i}) = \mathcal{N} \left(z_{n,j}; \mu_{n,j}+\gamma_{n,ij}\odot\sigma_{n,j}\odot\sigma_{n,i}^{-1}\odot(z_{n,i}-\mu_{n,i}), \sigma_{n,j} \odot \sqrt{1-\gamma_{n,ij}^2} \right)$
> >
> > Are there any quantities in the right-hand side of the above equation that do not depend on $n$?
> >
> > $\\{\gamma_{n,ij}\\}\_{(i,j) \in \mathcal{E}} = f\_{\boldsymbol{\phi}}(\boldsymbol{x}\_n)$
> >
> > Is it correct to consider $f_{\boldsymbol{\phi}}(\cdot)$ as a vector-valued function of dimension $|\mathcal{E}|$?
> >
> > Finally, in the traditional mean-field approximation in the experiments of Section 4.1, is it correct that independence both between instances and between dimensions are assumed?

---

> > > ### Author Response · Authors · 2024-08-12
> > >
> > > Yes, your understanding is fully correct.

---

> > > > ### Comment · Reviewer_BYBL · 2024-08-13
> > > >
> > > > Thank you for providing multiple responses to my questions.
> > > > I believe I now have an accurate understanding of the experiments in Section 4.1.
> > > >
> > > > Regrettably, I have to sαy the current manuscript lacks sufficient explanation. Moreover, providing a detailed explanation of the formulation for capturing inter-dimensional correlations in an appendix or elsewhere may not fully solve this issue. Capturing inter-dimensional correlations is not the main focus of this paper, so doing so might be misleading. In other words, it seems that the purpose and design of the experiments in Section 4.1 were not sufficiently aligned with the main theme of this paper.
> > > >
> > > > While I regret that the significant time you spent answering my questions has led to this outcome, I will have to lower the score by one point. I deeply appreciate your patience in responding to my inquiries.

---

### Official Review · Reviewer_ptS8 · 2024-07-13

**Soundness:** 2
**Presentation:** 2
**Contribution:** 2
**Rating:** 5
**Confidence:** 4

**Summary:**

The paper proposes to perform amortized variational inference, where the variational approximation has a tree-dependence structure across the instance-level latent variables. For this, the scale is decomposed into variance and correlation, where both are amortized through a neural network. A non-convex constrained optimization procedure is proposed to also learn the correlation structure, which ensures that the correlation structure forms an acyclic graph.

**Strengths:**

* In variational inference with structured variational families, learning the appropriate correlation structure from data is an important problem that hasn't been fully addressed.

**Weaknesses:**

* The proposed methodology is unclear at key places, making it hard to grasp exactly what is going on. For instance, it isn't clear to me what the constrained optimization problem in Eq 17 is supposed to do. Isn't an acyclic graph that spans *all* latent variables a solution to this problem? In that case, the resulting variational approximation is full-rank, which seems to defeat the whole point. Therefore, how exactly the authors obtain a sparse correlation structure needs to be clarified.
* A lot of the derivation in Section 2.1 is redundant. For instance, it isn't clear why the Cholesky factor has to be decomposed into variance and correlations involving a bunch of square roots. That is, the authors could have simply parameterized the $(i,j)$th element of the Cholesky factor as $L_{ij} = f_{\phi}\left(x_i, x_j\right)$. This would cut down the text of Section 2.1 by more than half since it is now obvious how to reparameterize. In fact, it is well known that manipulating the elements of the Cholesky factor is equivalent to manipulating the graph structure. (See Proposition 3 in [1] for example. I also suspect there should be a textbook result somewhere, but couldn't find such result.) This would also avoid all the square roots.
* Given the comment above, it is unclear *why* we have to take a tree structure among various different correlation graph structures.
* The paper is imprecise about the fact that it is solving a problem specific to *amortized variational inference* not just variational inference. In fact, a full-rank/dense scale matrix is strictly more expressive than any sparser graph approximation. Therefore, the proposed method does not make sense outside of the scalable amortized VI context. In that sense, the title and abstract are not specific enough about the setting and should be changed.
    * Line 31-36 "... struggling to alleviate the independence structure adopted by standard variational inference ...": We knew how to alleviate independence structure since the work of [2,3]. The statement is only true for instance level correlation in amortized VI.
    * Line 41: In "the study of variational inference with instance-level correlation remains under-explored," "the study of variational inference" should have been "the study of scalable amortized variational inference"

Overall, the crucial contribution of the paper seems to be about automatically learning the correlation structure, but it is unclear how this is done and how sparsity in the correlation structure is enforced. Furthermore, the motivation for tree structure (out of all structures) is unclear.

## Additional Comments
* Section 2.2: The additional Jensen gap can be avoided by using the sticking-the-landing gradient [13].
* While it is true that Gaussian mean-field families commonly used in deep latent variable models are restrictive, the recent development of MCMC-based augmentation methods [5,6,7,8] is able to overcome this problem.
* Line 55: Cite something for the ELBO. It is typical to cite [9,10].
* Line 28: The mean-field approximation was first proposed by [11] and popularized by [12].
* Line 38 "... share the same tree structure which is unrealistic in real applications": All variational inference is performing approximations. There is nothing realistic or realistic about any of the approximations. There are *accurate* or *less accurate* approximations.

## References
(Disclaimer: I am not the author of any of the papers below.)
1. Katzfuss, Matthias, and Joseph Guinness. "A general framework for Vecchia approximations of Gaussian processes." (2021): 124-141.
2. Hoffman, Matthew, and David Blei. "Stochastic structured variational inference." Artificial Intelligence and Statistics. PMLR, 2015.
3. Titsias, Michalis, and Miguel Lázaro-Gredilla. "Doubly stochastic variational Bayes for non-conjugate inference." International conference on machine learning. PMLR, 2014.
4. Kucukelbir, Alp, et al. "Automatic differentiation variational inference." Journal of machine learning research 18.14 (2017): 1-45.
5. Caterini, Anthony L., Arnaud Doucet, and Dino Sejdinovic. "Hamiltonian variational auto-encoder." Advances in Neural Information Processing Systems 31 (2018).
6. Geffner, Tomas, and Justin Domke. "MCMC variational inference via uncorrected Hamiltonian annealing." Advances in Neural Information Processing Systems 34 (2021): 639-651.
7. Doucet, Arnaud, et al. "Score-based diffusion meets annealed importance sampling." Advances in Neural Information Processing Systems 35 (2022): 21482-21494.
8. Thin, Achille, et al. "Monte Carlo variational auto-encoders." International Conference on Machine Learning. PMLR, 2021.
9. Jordan, Michael I., et al. "An introduction to variational methods for graphical models." Machine learning 37 (1999): 183-233.
10. Blei, David M., Alp Kucukelbir, and Jon D. McAuliffe. "Variational inference: A review for statisticians." Journal of the American statistical Association 112.518 (2017): 859-877.
11. Peterson, Carsten, and Eric Hartman. "Explorations of the mean field theory learning algorithm." Neural Networks 2.6 (1989): 475-494.
12. Hinton, Geoffrey E., and Drew Van Camp. "Keeping the neural networks simple by minimizing the description length of the weights." Proceedings of the sixth annual conference on Computational learning theory. 1993.
13. Roeder, Geoffrey, Yuhuai Wu, and David K. Duvenaud. "Sticking the landing: Simple, lower-variance gradient estimators for variational inference." Advances in Neural Information Processing Systems 30 (2017).

**Questions:**

n/a

---

> ### Author Rebuttal · Authors · 2024-08-07
>
> We appreciate your detailed comments, but we believe that you misunderstood our paper deeply. We hope our clarifications could help you recognize our contributions correctly.
>
> You mistakenly think that our main contribution is simply to Cholesky decompose the correlation matrix $\mathbf{R}$ as $\mathbf{R}=\mathbf{L}\mathbf{L}^\top$ and then re-parameterize the elements $\ell_{ij}$ in the lower-triangular matrix $\mathbf{L}$ by a neural network $f_\phi(\cdot)$, that is, $\ell_{ij}=f _ \phi(\mathbf{x} _ i, \mathbf{x} _ j)$, with $\mathbf{x} _ i$ denoting the $i$-th data instance. We want to emphasize that this is not our contribution. If this approach is adopted, we have to re-parameterize as many as $N(N+1)/2$ elements, i.e., all elements from the lower-triangular positions of $\mathbf{L}$ need to be re-parameterized, where $N$ is the number of instances in training dataset. That means we need to run the neural network $f_\phi(\cdot, \cdot)$ by $O(N^2)$ times for every epoch, which is computationally unacceptable, especially considering that $N$ could be as large as **millions** in practice.
>
>
> **Our main contribution lies at finding a way to reduce the required times of running the neural network $f_\phi(\cdot, \cdot)$ from $O(N^2)$ to $O(N)$**, making it applicable to scenarios even with millions of instances. To achieve this goal, we propose to restrict the correlation matrix $\mathbf{R}$ to a special form constructed from a tree $\mathbb{T}(\mathcal{V}, \mathcal{E})$. It should be pointed out that although $\mathbf{R}$ is constructed from a tree, it doesn't represent a tree. Actually, it is a dense matrix, with its $(i,j)$-th element $[\mathbf{R}] _ {ij}$ for $(i,j)\notin \mathcal{E}$ set as $\prod_{(s,t)\in \mathbb{P} _ {i\to j}}\gamma_{st}\triangleq \tilde \gamma_{ij}$.
>
> We prove that under the restricted correlation matrix $\mathbf{R}$, the lower-triangular matrix $\mathbf{L}$ possesses a very elegant form, as shown in Eq. (11) in the paper. The elegance lies at that although $\mathbf{L}$ still has $O(N^2)$ non-zero elements, all of these non-zero elements can be explicitly computed from the $|\mathcal{E}|$ parameters $\\{\gamma_{ij}\\} _ {(i,j)\in \mathcal{E}}$. For instance, the $(1, 4)$-th element in Eq.(11) can be computed as $\prod_{(s,t)\in \mathbb{P} _ {1\to 4}}\gamma_{st}$, where $\mathbb{P} _ {1\to 4}$ denotes the path conneting node 1 and 4 in the tree ${\mathbb{T}}$. The importance of this conclusion is that now we only need to re-parameterize the parameters $\\{\gamma_{ij}\\} _ {(i,j)\in \mathcal{E}}$, while computing the other non-zero elements in $\mathbf{L}$ with $\\{\gamma_{ij}\\} _ {(i,j)\in \mathcal{E}}$. In this way, we only need to re-parameterize $|\mathcal{E}|$ parameters, and thus only need to run $|\mathcal{E}|$ times of neural network $f_\phi(\cdot, \cdot)$, instead of $O(N^2)$ times in the vanilla method. For a tree, the number of edges $|\mathcal{E}|\le N -1$, thus we only need to additionally run $O(N)$ times of neural networks $f_\phi(\cdot, \cdot)$, in addition to the runs required by the mean-field method.
>
> For a mean-field variational inference that assumes independence among instances, for each epoch, it also need to run $O(N)$ times of neural network. Thus, the complexity of our proposed method is roughly only 2 times of the mean-field method. This is consistent with the elapsed time observed in the experiments.
>
>
>
>
> Below we respond to each point raised by the reviewer:
>
> **Q1: A lot of the derivation in Section 2.1 is redundant. The authors could have simply parameterized the Cholesky factors.**
>
> A1: As discussed above, directly parameterizing the Cholesky factors is computationally consuming, which requires to run the neural network $f_\phi(\cdot, \cdot)$ by $O(N^2)$ times. Section 2.1 is mainly devoted to how to achieve a correlation-capturing variational inference method that only requires running the neural network by $O(N)$ times.
>
>
> **Q2: Why to take a tree structure among various different correlation graph structures.**
>
> A2: As elaborated above, by restricting the correlation matrix ${\mathbf{R}}$ to a form that is constructed from a tree, we can prove that all of the non-zero elements in ${\mathbf{L}}$ can be explicitly computed from the $|{\mathcal{E}}|$ parameters $\gamma_{ij}$ for $(i,j)\in {\mathcal{E}}$. Then, we only need to re-parameterize the $|{\mathcal{E}}|$ parameters $\gamma_{ij}$, instead of all the non-zero elements in ${\mathbf{L}}$. That's why we have to restrict the correlation matrix ${\mathbf{R}}$ to be constructed from a tree. Again, we want to point out that ${\mathbf{R}}$ is only constructed from a tree, but it is a dense matrix.
>
>
> **Q3: What the constrained optimization problem in Eq 17 is supposed to do.**
>
> A3: As discussed above, we need to construct a tree to learn the instance-level correlations. Besides trivial methods like heuristic methods and greedy search algorithm, it can also be learnt from data by our constrained optimization. Specifically, we use a real matrix to represent the correlation parameters, which satisfies the acyclicity constraint if and only if the indicator function $h(\mathbf{A}) = 0$ as shown in Eq. 17. This allows us to formulate a constrained optimization problem with a smooth equality constraint, which can be solved with Lagrangian multipliers by incorporating a penalty term for the indicator function.
>
> **Q4: Imprecise about amortized variational inference not just variational inference.**
>
> A4: We apologize for the impreciseness brought by our terminologies, and we will reflect this in our title of the final version.
>
> **Q5: Sticking-the-landing gradient and MCMC-based augmentation methods.**
>
> A5: We thank the reviewer for pointing out some feasible alternatives. However, the MCMC-based augmentation methods may cause inefficiency by introducing time-consuming sampling procedure.

---

> > ### Comment · Reviewer_ptS8 · 2024-08-10
> > **Response**
> >
> > Thank you for the response.
> >
> > > We appreciate your detailed comments, but we believe that you misunderstood our paper deeply. We hope our clarifications could help you recognize our contributions correctly.
> >
> > Now, I believe I better understand the contribution. Thank you for the explanation. However, there are some clarity issues, and I believe the writing could be substantially improved. Let me be more precise. In Section 2.1, which is crucial to understanding the contribution of the paper, there are zero citations. Therefore, it is somewhat hard to distinguish which part of the claims are mathematical facts that are original to this paper, approximations, or well-known results. For instance, if I understood correctly, Eq 3 is saying we can retrieve the full correlation structure just from the tree right? What is unclear here is whether all fully-correlated Guassians can be retrieved from this formula or a subset of correlated Gaussians. If this is a known fact in graphical models, a citation would be needed here.
> >
> > Furthermore, the main motivation for using trees here is that one can model $N^2$ interactions from $N$ queries to a neural network, which is nowhere mentioned in the introduction! Both the abstract and the introduction only mention that the method proposed here is more "scalable" and can be "reparameterized efficiently." However, the derivation of the reparameterization trick itself appears fairly straightforward and, in my opinion, is not the most critical contribution here. Basically, if one doesn't carefully read Lines 133-135, it is easy to miss the fundamental motivation for the tree structure.
> >
> > With that said the role of the optimization problem is now clear. I do agree that the paper has some original contributions, so I am happy to raise my score. However, as other reviewers pointed out and pointed out above, the paper has some clarity issues and I feel that a thorough revision and an additional round of review would be very beneficial to the paper.

---

> > > ### Author Response · Authors · 2024-08-11
> > >
> > > We appreciate the reviewer for carefully reading our explanations and recognizing our original contributions in this paper. As for the clarity issue, we agree that due to the limited space, we haven't provided sufficient  explanation to the computational complexity as well as the elegance of the proposed method. But as an additional page is allowed in the final version, we believe more detailed explanations on the complexity and the distinguished properties of the proposed method, as stated in our response to you above, could be easily incorporated into the paper. Thus, we think this issue could be easily addressed in the final version.
> > >
> > > As for the question on computing the correlation coefficients from a tree (Eq. (3)), yes, the proposed correlation matrix ${\mathbf{R}}$ constructed from a tree can only represent a subset of all legitimate correlation matrices. We will make this point more clear and cite appropriate literatures here.

---

### Author Rebuttal · Authors · 2024-08-07

We thank the reviewers for their careful reading and detailed feedback.

Please refer to the attached PDF for new results. We note that we have provided additional insights about our learnt tree structure obtained through training. In Figure 1, we plot the tree structure learnt over constrained clustering task on the MNIST dataset, showing tight connection between our learnt correlation structure to the category distribution. Further, we supplement the constrained clustering results with both means and standard deviations for stability analysis as shown in Table 1.

---

### Decision · Program_Chairs · 2024-09-25

**Decision:**

Accept (poster)

**Comment:**

The paper presents a method to perform amortized VI with a distribution family in between mean-field and fully connected Gaussians. Instead, the authors proposed to use a tree-structured Gaussian graphical model over data instances. Reviewers found the paper to be interesting, novel, and sound and with experiments that supported the main claims. There was considerable back-and-forth during the rebuttal and author-reviewer discussion phase. Much of it surrounding technical clarification of the methods, with the result that two reviewers raised their ratings and one lowered their rating. In the end, all reviewers appreciated the contributions and unanimously recommended accept. The most significant issue that can be addressed to improve the paper is to resolve potential ambiguity surrounding the problem setting and motivation. For example, the authors are encouraged to address the post-rebuttal comments of Reviewer ptS8 about highlighting more clearly the main motivation of using a tree structure, as well as the post-rebuttal comments by Reviewer BYBL about the discrepancy of using an instance-level tree in one of the experiments.